# *Sargassum* Seaweed as a Source of Anti-Inflammatory Substances and the Potential Insight of the Tropical Species: A Review

**DOI:** 10.3390/md17100590

**Published:** 2019-10-17

**Authors:** Puspo Edi Giriwono, Diah Iskandriati, Chin Ping Tan, Nuri Andarwulan

**Affiliations:** 1Department of Food Science and Technology, Faculty of Agricultural Engineering and Technology, Bogor Agricultural University, Bogor 16680, Indonesia; ginasaraswati@gmail.com (S.); pegiriwono@apps.ipb.ac.id (P.E.G.); 2Southeast Asian Food and Agricultural Science Technology (SEAFAST) Center, Bogor Agricultural University, Bogor 16680, Indonesia; 3Primate Research Center, Bogor Agricultural University, Bogor 16151, Indonesia; atie@indo.net.id; 4Department of Food Technology, Faculty of Food Science and Technology, Universiti Putra Malaysia, Serdang 43400, Malaysia

**Keywords:** bioactive compounds, inflammation, mechanism, *Sargassum*, tropical

## Abstract

*Sargassum* is recognized both empirically and scientifically as a potential anti-inflammatory agent. Inflammation is an important response in the body that helps to overcome various challenges to body homeostasis such as microbial infections, tissue stress, and certain injuries. Excessive and uncontrolled inflammatory conditions can affect the pathogenesis of various diseases. This review aims to explore the potential of *Sargassum*’s anti-inflammatory activity, not only in crude extracts but also in sulfated polysaccharides and purified compounds. The tropical region has a promising availability of *Sargassum* biomass because its climate allows for the optimal growth of seaweed throughout the year. This is important for its commercial utilization as functional ingredients for both food and non-food applications. To the best of our knowledge, studies related to *Sargassum*’s anti-inflammatory activity are still dominated by subtropical species. Studies on tropical *Sargassum* are mainly focused on the polysaccharides group, though there are some other potentially bioactive compounds such as polyphenols, terpenoids, fucoxanthin, fatty acids and their derivatives, typical polar lipids, and other groups. Information on the modulation mechanism of *Sargassum*’s bioactive compounds on the inflammatory response is also discussed here, but specific mechanisms related to the interaction between bioactive compounds and targets in cells still need to be further studied.

## 1. Introduction

Inflammation is known to be a protective strategy that evolves in high-level organisms in response to threats interfering with body homeostasis. Types of threats causing inflammation include microbial infections, tissue stress, and certain injuries [1,2]. The well-known symptoms of classic inflammation are redness, pain, swelling, and fever [3]. Recent literature shows that inflammation operates via an advanced system and has a broad impact on various physiological aspects and human pathology. Although inflammatory response plays a pivotal role in protecting cellular physiological conditions, this process is only needed for a short time to avoid the stages of escalation that result in undesirable conditions, such as collateral tissue damage [1]. Uncontrolled and excessive inflammatory conditions can lead to various health problems, for example, multiple sclerosis, cancer, arthritis, atherosclerosis, heart disease, obesity, dermatitis, migraine, irritable bowel disease, insulin resistance, autoimmune, and other diseases [4].

An inflammatory response begins with the recognition of various stimuli by a cellular transmembrane receptor. This condition triggers the transactivation of several important transcription factors, primarily nuclear factor kB (NF-kB), which are responsible for the regulation of various genes related to inflammation. In the unstimulated condition, the NF-kB protein is located in the cytoplasm and bound to a protein inhibitor IkB. The degradation of IkB due to upstream inflammatory signals allows NF-kB to undergo nuclear translocation and regulates the downstream inflammatory response by binding to the kB site in the DNA’s structure. This well-known inflammatory signaling is commonly called a canonical pathway. In addition to NF-kB activation, inflammatory stimulation can lead to the activation of MAPK (mitogen-activated protein kinase) pathways, such as ERK (extracellular signal-regulated kinases), p38 MAPK, and JNK (cJun NH_2_-terminal kinases) [5]. These three protein kinases also play a role in the regulation of genes associated with inflammation through the transactivation of AP-1 proteins. The inflammatory response is generally characterized by the overproduction of prostaglandin E_2_ (PGE_2_), nitric oxide (NO), pro-inflammatory cytokines (such as tumor necrosis factor (TNF)-α, interleukin (IL)-6, and IL-1β), and increased production of reactive oxygen species (ROS) [6]. The overproduction of NO and PGE_2_ are concomitant with the increased activity of inducible nitric oxide synthase (iNOS) and cyclooxygenase-2 (COX-2), respectively. The continuous energy surplus usually occurring in metabolic syndromes can trigger unresolved low-grade chronic inflammation, which may lead to various conditions such as increased oxidative stress, metabolic tissue damage, and/or insulin resistance [7].

Brown seaweed, especially *Sargassum*, is known to have many health benefits. Fitton [8] concluded that the East Asian lifestyle of using brown seaweed as a part of the staple diet is associated with a low incidence of cancer in the region. Oh et al. [9] reported that the consumption of whole brown seaweeds (including *Sargassum fulvellum* and *S. fusiforme*) contributed to ameliorating systemic inflammation and insulin resistance in high fat diet-induced obese mice. Husni et al. [10] also revealed that a diet supplemented by whole seaweed powder (*S. hystrix*) improved stress-induced liver inflammation conditions in Wistar rats. Based on information in Donguibogam (Korean-Oriental Medical Textbook), seaweeds such as *S*. *fulvellum* and *S. thunbergii* have been used for generations to treat edema and painful scrotums [11]. Chinese people use various species of *Sargassum* to treat scrofula, edema, arteriosclerosis, skin diseases, hypertension conditions, liver organ swelling, neurosis, angina pectoris, esophagitis, and chronic bronchitis bronchitis [12]. Studies on utilizing *Sargassum* as an anti-inflammatory agent are very interesting because this is associated with the empirical application of *Sargassum* for acute and chronic inflammation. Currently, intensive screening of anti-inflammatory agents from natural ingredients is being undertaken to determine alternatives to established synthetic drugs and is expected to show decreased side effects in humans with inflammatory disorders, particularly chronic inflammation [4,11,13].

This review aims to explore the potential of *Sargassum* brown seaweeds as a source of anti-inflammatory agents and their modulation mechanisms. Since different latitudes (tropical and subtropical areas) will greatly affect brown seaweed’s bioactive components [14,15], special insight on the potential use of tropical *Sargassum* as an anti-inflammatory agent is discussed here. The tropical region has a promising availability of *Sargassum* biomass because its climate allows for optimal seaweed growth throughout the year. This climate condition is expected to have an important implication for the commercial exploration of tropical *Sargassum* as a source of functional ingredients.

## 2. *Sargassum* Species Are a Source of Anti-Inflammatory Agents

According to the literature summary of *Sargassum*’s anti-inflammatory activity (2005–2019) shown in Table 1, Table 2, Table 3 and Table 4, the samples tested were generally divided into three major groups: Crude extracts and their partitions, crude sulfated polysaccharides, and purified compounds from multilevel fractionation. Various *Sargassum* species used in several studies of anti-inflammatory activity originated from two subgenera of *Sargassum*, namely *Bactrophycus* and *Sargassum*.

The *Sargassum* genus is generally divided into four subgenera: *Phyllotrichia, Bactrophycus, Arthrophycus,* and *Sargassum.* This subgenus classification is based on the morphological characteristics of the seaweed thallus. In addition, the distribution pattern can also be used to distinguish between *Sargassum* subgenera. The most common subgenera found in subtropical/temperate regions are *Bactrophycus* and *Arthrophycus.* Subgenus *Phyllotrichia* is only found in Australia and adjacent areas, while the *Sargassum* subgenus is widely distributed in tropical areas [16]. Due to the high level of morphological plasticity caused by differences in environmental condition, molecular marker techniques combined with morphological observation are implemented to resolve taxonomic issues [17]. A sequence of the internal transcribed spacer of nuclear ribosomal DNA (ITS) is commonly used to analyze the phylogenetic relationship among *Sargassum* species. Figure 1 shows the phylogenetic tree of the *Sargassum* genus based on internal transcribed spacer (ITS)-2 gene sequences. The phylogenetic tree was constructed based on several *Sargassum* species used in various anti-inflammatory activity studies. The accession number of each gene sequences is obtained from the database of the National Center for Biotechnology Information.

### 2.1. Crude Extracts and Their Partitions

An anti-inflammatory activity summary of *Sargassum*’s crude extracts and their partitions is shown in Table 1. Various types of solvents with different polarities are used for extraction and partitioning of the anti-inflammatory compounds. Mun et al. [18] compared the anti-inflammatory activity of the ethanolic extracts from fermented and non-fermented (fresh) samples of the brown seaweed *S*. *thunbergii* in lipopolysaccharide (LPS)-stimulated RAW 264.7 macrophage cells. Fermentation was initiated in fresh S. *thunbergii* by inoculation of kimchii-isolated *Lactobacillus* sp. SH-1. The results showed that fermentation increased the anti-inflammatory effect as indicated by a decreased production of NO and a suppression of iNOS and COX-2 expression. Partitioning of the ethanolic extract resulted in stronger anti-inflammatory activity localized in the relatively lipophilic fractions (85% methanol and n-hexane fraction obtained from the chloroform phase). The n-butanol fraction obtained from the water phase exhibited stronger anti-inflammatory activity than the water fraction, which was proven by a decrease in NO production, a suppression of protein and mRNA expression of IL-1β, IL-6, TNF-α, iNOS, and COX-2, and inhibition of MAPKs phosphorylation, especially the JNK pathway. These facts indicated that the relatively lipophilic compounds tended to be responsible for the anti-inflammatory activity. This was supported by several other studies showing a similar tendency that a relatively lipophylic extract or fraction has a promising anti-inflammatory effect [19,20,21,22,23,24,25,26,27,28,29,30,31,32,33].

Kang et al. [22] reported that the dichloromethane fraction of *S*. *fulvellum* and ethanol fraction of *S*. *thunbergii* had a stronger effect in inhibiting mouse ear edema than the water fraction. Based on the GC–MS analysis result of both mentioned fractions, compounds associated with fatty acids and simple organic compounds, such as hexadecanoic acid, neophytadiene, tetradecanoic, 8-heptadecene, and 3,7,11,15-tetramethyl-2-hexadecen-1-ol, dominated the two fractions content. These compounds were thought to be the responsible anti-inflammatory agent. Molecular docking study conducted by Balachandran et al. [93] found that hexadecanoic acid and (E)-9-octadecenoic ethyl ester contained in *S*. *wightii* were effective in inhibiting the COX-2 enzyme activity. Several other compounds that are allegedly responsible for the anti-inflammatory activity of the non-polar fraction include fucoxanthin or its derivatives [33], fucosterol or other steroid compounds [20,23,28,29,31,33,51], and phenolic compounds [18,23,30,31,33]. Although phenolic compounds are commonly known to have hydrophilic natures, their polarities will actually depend on their chemical structures. A phenolic component consists of various groups with a wide range of structural variations [94]. Various types of lipophilic phenolic components can be found in nature in the form of steryl phenolic acid esters, phenolic acid lactones, phenolic fatty acid esters, and others [95].

In addition to the promising potential of lipophilic extracts/fractions, several studies had shown that a water-soluble extract of *Sargassum* was also effective in suppressing the inflammatory response [35,44,48,71,81]. Kang et al. [71] reported that the *S. fusiforme*’s water extract could suppress the production of TNF-α in the LPS-induced C2C12 myotube cells and increase the production of cytokines associated with increased insulin sensitivity, such as IL-6 and IL-10. The chemical profile of that water extract was not identified, but the phenolic component was thought to be partially responsible for its anti-inflammatory activity. Jaswir et al. [35] showed that water extracts from several species of *Sargassum* had an NO inhibitory effect in LPS-induced RAW 267.4 cells, even though the anti-inflammatory activity was not as strong as that of other water extracts obtained from *Padina australis* and *Turbinaria turbinata*. They found that the anti-inflammatory activity of the observed brown seaweed water extract was positively correlated with fucose and uronic acid content. Another study showed that the polar β-glucan extract obtained from *S. crassifolium* has a potent anti-inflammatory activity in the rat ear edema model [48].

*Sargassum*’s crude extract not only shows its effectiveness in the acute inflammatory model but also in the chronic inflammatory model, as reported by several studies [28,29,44,50]. The study of Dhas et al. [44] reported that the administration of gold nanoparticles made from a *S*. *swartzii* water extract (28 days treatment) improved insulin sensitivity and reduced the serum production of TNF-α, IL- 6, and CRP in alloxan-induced diabetic rats. The administration of the methanol extract of *S. subrepandum* (four months, 100 mg/kg body weight) in rats with dyslipidemia triggered an improvement in their plasma lipid profile and reduced the content of MDA, NO, leptin, and TNF-α in rat serum [50]. Dyslipidemia in the mentioned study was triggered by the atherogenic diet given to Sprague Dawley rats for eight consecutive months. Kwon et al. [28] and Gwon et al. [29] proved that the meroterpenoid-rich fraction obtained from ethanolic extract of *S*. *serratifolium* improved the inflammation condition in both high-fat diet-induced and high-cholesterol diet-induced C57BL/6J mice.

### 2.2. Crude Sulfated Polysaccharides

Studies on the anti-inflammatory activity of the *Sargassum* crude sulfated polysaccharides (CSP) have been carried out by several research groups [39,96,97,98,99,100,101,102,103,104,105,106,107] and are shown in Table 2. Sulfated polysaccharides (SP), such as fucoidan, carrageenan, and ulvan, are recognized as the typical functional compounds derived from various types of marine algae. Fucoidan is a dominant sulfated polysaccharide found in brown seaweed.

Sanjeewa et al. [100] reported that CSP yielded from Celluclast enzyme digestion against *S*. *horneri* had the same Fourier transform infrared (FT-IR) spectrum as a commercial fucoidan. Wen et al. [102] found that the purified SP derived from *S*. *horneri* (fractionated by column Q Sepharose Fast Flow) contained higher fucose and sulfate ester content than its crude form, and this ester exists along with the elevated anti-inflammatory activity.

The molecular weight (Mw) of SP seems to affect its biological activity. A study of Neelakandan and Venkatesan [101] revealed that a 10 kDa fraction derived from the CSP of *S*. *wightii* exhibited stronger anti-inflammatory activity than the fraction with Mw > 10 kDa and was more effective than its crude extract. The anti-inflammatory effect of their study was demonstrated by the inhibition of edema formation in carrageenan-induced rat paw edema, the suppression of leukocyte and neutrophil migration in the peritonitis model, and inhibition of edema formation in the Freund’s complete adjuvant-induced arthritis model. The result of their study controverts the findings of Sanjeewa et al. [96], who reported that the SP fraction with Mw > 30 kDa was the most effective in suppressing NO production (IC50 = 87.12 μg/mL) by LPS-induced RAW 264.7 cells among other lower Mw fractions (<5, 5–10, and 10–30 kDa). Wu et al. [103] found that the SP fraction with an Mw of 386.1 kDa had the highest NO inhibitory activity compared to other fractions (1193.2, 864.4, 106.3, 55.9, 15.4, and 1.9 kDa). Fractions with a high Mw (1193.2, 864.4, and 386.1 kDa) tended to have higher anti-inflammatory effects than low Mw fractions (55.9, 15.4, and 1.9 kDa). The effect of SP’s molecular weight on anti-inflammatory activity will depend on various factors, i.e., the testing model, sample, extraction method, and SP structural configuration, including its active groups. The effect of sulfation levels on SP’s anti-inflammatory activity was also studied by Wu et al. [103], who found that the 386.1 kDa fraction with 9.42% sulfate content showed the highest NO inhibition among other samples with different sulfate content (50.83%, 31.01%, and 0.84%). The SP fraction with 50.83% sulfate content showed the lowest NO inhibitory effect. Although the study of Wen et al. [102] stated that an increase in sulfate esters in the SP coincided with an enhancement of the anti-inflammatory effect, the study of Wu et al. [103] confirmed that the correlation between them is not always linear.

Fernando et al. [4] summarized that fucoidan functionality will depend on the monosaccharide sequences, sulfation levels, and connectivity of sulfate groups. Fucoidan has a similar characteristic to heparan sulfate, which has an antithrombotic effect with a low bleeding effect [114]. Fucoidan has been reported to inhibit the secretion of gelatinase A and stromelysin in IL-1β-induced fibroblasts cells, increase the association of MMPs (matrix metalloproteinases) with their inhibitors (TIMPs), and decrease the activity of leukocyte-secreted elastase. Based on these facts, fucoidan can be used to overcome aberrant extracellular matrix degradation, which usually occurs in inflammatory-related disease. The fucose-containing sulfated polysaccharide (FCSP) term is now considered more relevant than fucoidan because the sugar monomer constituent of common bioactive SP in brown seaweed is not only fucose, but also other sugars such as galactose, rhamnose, mannose, xylose, and glucose [115].

### 2.3. Purified Bioactive Compounds

Some purified compounds of *Sargassum*, which have been tested for anti-inflammatory activity, include: (1) Terpenoid compounds, such as tuberatolide B, loliolide, sargachromenol, sargachromanol D, sargachromanol G, sargaquinoic acid, sargahydroquinoic acid, isoketochabrolic acid/IKCA, isonahocol E3, and fucosterol fucosterol [19,26,116,117,118,119,120,121,122,123,124,125,126,127,128,129]; (2) fucoxanthin and apo-9’-fucoxanthinone [27,117,130,131,132,133]; (3) the polysaccharide group, including alginic acid and pure FCSPs [134,135,136,137,138,139,140,141]; (4) phenolic compounds (phlorotannins) [142,143]; and (5) other groups, such as aryl polyketide lactones and grasshopper ketones [65,144,145]. A summary can be seen in Table 3 and Table 4. Carotenoid compounds, such as fucoxanthin and apo-fucoxanthinone, can also be categorized into the terpenoid group because of the presence of tetraterpenoid derivatives with eight isoprene units [146]. In addition to these compounds, there are other anti-inflammatory compounds contained in brown seaweed: (1) Fatty acids and lipid derivatives such as stearidonic acid (SDA), timnodonic acid, eicosapentaenoic acid (EPA), hexadecanoic acid, 9-octadecenoic ethyl ester, and 7-methoxy-9-methylhexadeca-4,8-dienoic acid and (2) other groups, such as γ-aminobutyric acid, methyl salicylate, benzoic acid, 2-hydroxyethyl ester, diethyl phthalate, pheophorbide A, and pheophytin [4].

According to the above summary, there are still many potential anti-inflammatory compounds from the *Sargassum* species to be unraveled, especially groups of fatty acids and lipid derivatives. Some *Sargassum* species are reported to be rich in anti-inflammatory fatty acids, primarily EPA (C20:5n3) and SDA (C18:4n3) [14,148]. Chen et al. [148] revealed that some *Sargassum* species often used in traditional Chinese medicine have satisfactory ratios of polyunsaturated fatty acid (PUFA) to saturated fatty acid (SFA; >0.45) and n-6/n-3 (<10), which can improve pathological conditions in cardiovascular diseases, which are strongly related to inflammatory disorders. Some essential PUFAs in *Sargassum*, especially EPA and SDA, are incorporated into glycolipid structures [149,150,151]. PUFA-rich polar lipids in *Sargassum*, including the glycolipid group, are reported to have biological activities that benefit human health, such as fibrinolytic and anti-cancer activities [149,152]. The NMI fatty acid (non-methylene interrupted, C20:2n11,16) found in *S. marginatum* is thought to correlate with apoptotic activity in human blood cancer cells (HL-60) [149].

#### 2.3.1. Terpenoids

Terpenoids are a large group of secondary metabolites. This compound is composed of isoprene units (2-methyl-1,3 butadiene, C_5_H_8_) and produced by the biosynthesis process with mevalonate as a parent [153]. Sterol is a triterpenoid derivative containing six units of isoprene, and fucosterol is a dominant sterol commonly found in brown seaweed [4]. Several mentioned compounds such as tuberatolide B, sargachromenol, sargachromanol D, sargaquinoic acid, sargahydroquinoic acid, and isonahocol E3 are included in the meroterpenoid group. The group of meroterpenoid is characterized to possess a polyprenyl structure bound to hydroquinone or similar aromatic rings [12]. Isoketochabrolic acid (IKCA) belongs to the C18 terpenoid-related carboxylic acids group [154]. Loliolide or 6-hydroxy-4,4,7a-trimethyl-5,6,7,7a-tetrahydrobenzofuran-2(4H)-one (HTT) is included to the monoterpenoid hydroxylactone group [155]. Various terpenoid compounds have been shown to play an important role as modulators in the NF-kB signaling pathway, so this group of compounds is known as a potent anti-inflammatory agent [153]. The non-polar characteristic of terpenoids will make them diffuse into inflamed cells easily, and may produce a rapid effect in modulating the inflammatory response. Chemical structures of several terpenoid compounds can be seen in Figure 2.

#### 2.3.2. Fucoxanthin and Its Derivatives

Fucoxanthin is the dominant carotenoid in brown seaweed, which can absorb the light spectrum in the range of 450–540 nm. Fucoxanthin has a unique structure, which includes allenic bonds, conjugated carbonyl, epoxides, and acetyl groups. Carotenoids are usually found in nature as trans isomers. However, cis-isomers can still be found in small amounts due to the isomerization process. Several studies have shown that the cis–trans configuration of a carotenoid isomer affects its bioactivity [117,163,164]. Heo et al. [117] showed that all-trans-(6’R) fucoxanthin isomer more effectively inhibited NO and TNF-α production in LPS-stimulated RAW 264.7 cells than its cis isomers (9’-cis-(6’R) fucoxanthin and a complex of 13-cis (6’R) fucoxanthin and 13’-cis- (6’R) fucoxanthin). A complex of the 13-cis and 13’-cis- (6’R) fucoxanthin isomer showed significant cytotoxicity in cells, so the anti-inflammatory effect can be associated with its cytotoxicity. Apo-9-fucoxanthinone, a product of fucoxanthin degradation, is also reported to have promising anti-inflammatory activity [27,130,131,132,133]. Chemical structures of fucoxanthin and its derivatives are shown in Figure 3.

#### 2.3.3. Other Lipid-Soluble Compounds

The other types of lipid-soluble anti-inflammatory compounds that have been reported are polyketide macrolactone and grasshopper ketone. Chemical structures of those compounds can be seen in Figure 4. Polyketide is a secondary metabolite compound characterized by the presence of alternating carbonyl and methylene groups and is generally produced by fungi [167]. Maneesh et al. [144] succeeded in isolating two types of aryl-polyketide lactone compounds from *S*. *wightii.* These compounds had strong anti-inflammatory and antioxidant activity. The two isolated compounds are 4-(8-ethyl-tetrahydro-7-oxo-2*H*-pyran-5-yl)-propyl-4′-methyl benzoate (compound 1) and methyl-2-(12-oxo-7-phenyl-8-vinyl-1-oxa-4,9- cyclododecadien-3-yl)-acetate (compound 2). The study found that the isolated aryl polyketide lactones had a more effective 5-LOX and COX-2 inhibitory activity than standard synthetic drugs, such as ibuprofen and sodium salicylate, and had higher COX-2/COX-1 selectivity than aspirin and ibuprofen. The inhibitory activity of the aforementioned compounds against 5-LOX, COX-2, and COX-1 enzymes is strongly influenced by their electronic and hydrophobic characteristics. Electron-rich centers, such as the cyclic esters and vinyl or aryl substituents contained in aryl polyketide lactones, act as unsaturation centers determining the inhibition effect on pro-inflammatory enzymes.

The grasshopper ketone (GK) or 4-[(2*R*,4*S*)-2,4-dihydroxy-2,6,6-trimethylcyclohexylidene]but-3-en-2-one was successfully isolated from the hexane fraction of *S*. *fulvellum* by Kang et al. [65] and Kim et al. [145]. GK was found to significantly decrease cytokine production in concanavalin A-stimulated splenocytes BALB/c mice with no cytotoxicity [65]. This compound also significantly inhibited the production of iNOS, COX-2, and several pro-inflammatory cytokines in LPS-induced RAW 264.7 [145].

#### 2.3.4. Polysaccharides

The most abundant active polysaccharides derived from brown seaweed are FCSPs and alginic acid. Their chemical structures can be seen in Figure 5. The main constituent of FCSPs is L-fucopyranose, which may be substituted with sulfate or acetate and/or have side branches containing fucopyranoses or other glycosyl units [169]. Alginic acid is anionic polysaccharide consisting of β-D-mannuronate and α-L-guluronate as basic monomers. Alginic acid is reported to exhibit potential anti-inflammatory activity in the rat arthritis model induced by type-2 collagen and Freund’s complete adjuvant [138,139]. 

The majority of exploratory studies of pure compounds from *Sargassum* (Table 3) use single components as the only responsible agent in improving inflammatory conditions, except for the studies by Hwang et al. [137] and Lin et al. [141].Hwang et al. [137] found that the combination of LMF (low molecular weight fucoidan; with a Mw around 0.8 kDa) and HS-Fucox (high stability fucoxanthin) derived from *S*. *hemiphyllum* effectively suppressed inflammatory response in LPS-induced CaCo2 cells co-cultured with *Bifidobacterium lactis*. This was indicated by a decreased production of TNF-α and IL-1β, as well as increased production of anti-inflammatory cytokines such as IL-10 and IFN-γ. In addition, the treatment of LMF-HS Fucox also helped maintain intestinal epithelial cell integrity, which was indicated by an increase in the mRNA expression encoding the tight junction protein (occludin, claudin-1, and claudin-2). LMF can promote the growth of probiotic *B*. *Lactis*, thereby helping to maintain the structural integrity of colon cells. Park et al. [170] reported that LMF (Mw ± 1 kDa) has a stronger NO inhibition effect than the high molecular weight fucoidan/HMF (Mw ± 100 kDa) in the LPS-induced RAW 264.7 cells. In the model of type 2 collagen-induced arthritis, oral administration of HMF (300 mg/kg, 49 days) exacerbated the severity of arthritis and inflammatory conditions in joint cartilage, whereas LMF treatment reduced arthritis severity and the levels of Th1-dependent collagen-specific IgG_2a_. This contradicts the findings of Sanjeewa et al. [96] and Wu et al. [103], as discussed in the previous section. These two studies also observed the NO inhibition level in a similar cell culture model. Sanjeewa et al. [96] obtained an FCSP-rich fraction through filtration techniques, while Wu et al. [103] obtained crude FCSPs with different molecular weights using the acid hydrolysis technique (also as conducted by Park et al. [170]). This contradiction could not be further explained because the chemical characteristic information for each FCSP fractions is limited, especially for the sulfation level which may greatly determine the bioactivity of FCSPs [103,171].

#### 2.3.5. Phenolic Compounds

Phlorotannin is a typical phenolic group of brown seaweed. These compounds have a broad molecular size range (from 126 Da to 650 kDa) and are divided into six main classes, fucols, phloretols, fucophlorethols, fuhalols, isofuhalols, and eckols. Phlorotannin is composed of phloroglucinol units (1,3,5-trihydroxybenzene) with different degrees of polymerization. The classification of phlorotannin is based on the different bond types between their constituent units [143]. Lopes et al. [143] reported that the phlorotannin content in some types of seaweed belonging to the Sargassaceae family ranged from 74.96 to 815.82 mg phloroglucinol/kg. The main phlorotannin classes found in *Sargassum* are fuhalols, phlorethols, and fucophlorethols [173,174,175]. Li et al. [173] reported that the phlorotannin in S. *fusiforme* was dominated by a fuhalol group with DP of 2–12. Lopes et al. [143] proved that phlorotannin affected the NO levels in LPS-induced RAW 264.7 through NO direct scavenging and/or the modulation of inflammatory signals in cells. The anti-inflammatory activity of phlorotannin is strongly influenced by its qualitative composition. The chemical structures of phloroglucinol and examples of several phlorotannin classes commonly found in *Sargassum* are shown in Figure 6.

## 3. The Potency of Tropical *Sargassum* as an Anti-Inflammatory Agent

### 3.1. Latest Reports on Anti-Inflammatory Activity of Tropical *Sargassum*

The majority of *Sargassum* crude extracts tested on anti-inflammatory screening are derived from subtropical samples. Seventeen out of 73 studies used samples from tropical regions [34,35,36,39,40,41,42,43,44,45,47,48,49,51,52,53,54]. Most of the observed tropical species came from subgenus *Sargassum*, including *S. polycystum, S. wightii, S. swartzii, S. crassifolium, S. binderi*, and *S*. *ilicifolium.* Some subtropical species tested in the crude extract studies were dominated by the subgenus *Bactrophycus*, including *S. hemiphyllum, S. muticum, S. sagamianum, S. macrocarpum, S. micracanthum, S. coreanum, S. horneri, S. fusiforme, S. miyabei, S. serratifolium, S. fulvellum, S. confusum, S. siliquastrum, S. pallidum, S. ringgoldianum*, and *S*. *thunbergii*. However, some species belonging to the subgenus *Sargassum* can also be found in subtropical areas, such as *S. patens, S. wightii, S. vulgare, S. subrepandum*, and *S*. *swartzii* [23,37,38,46,50,55,56]. The differentiation between tropical and subtropical samples is based on the thorough evaluation of the sampling location or coordinates information provided in each study. Tropical samples originated from area near the equator (from 23.5° further north to 23.5° southern latitude). While subtropical samples originated from area between 23.5° and 66.5° north and south. Some models used in the screening studies of tropical *Sargassum*’s crude extracts are rat edema [41,42,45,48,49,51], the in vitro inhibition of 5-LOX, COX-2, and COX-1 enzymes [39], the stabilization of red blood cell (RBC) membrane [43], in vitro inhibition of albumin denaturation [40,43], and inhibition of proteinases [43]. These models are still unable to fully describe the comprehensive mechanisms of bioactive compounds in inhibiting the inflammatory process. Some studies tried to screen the anti-inflammatory activity of several tropical *Sargassum* varieties using the RAW 264.7 cell model, but the available mechanism information is only limited to the inhibition of NO production after LPS stimulation [35,36,47,54]. Moni et al. [34], Dhas et al. [44], and Motshakeri et al. [53] successfully proved that *Sargassum* extract could improve inflammatory conditions in the diabetic rat model, but information about cellular mechanisms is still not obvious.

There are few researchers still using a protein denaturation inhibition model to screen anti-inflammatory activity in this new prostaglandin era. The reason behind choosing this model is based on the parallelism between two physio-pathological phenomena, namely inflammation and protein denaturation, both of which can be caused by the same stimulants (e.g., heat, radiation, organic solvents, etc.) [176]. Various types of NSAIDs (non-steroidal anti-inflammatory drugs) are reported to inhibit the in vitro denaturation of biologically active plasma proteins [177]. Protein denaturation can be the initial stage of further protein modification—for instance, protein glycosylation. Abnormal protein glycosylation phenomena occur frequently in chronic inflammatory diseases [178]. Nevertheless, the protein denaturation model is less convincing for most researchers today because the protein denaturation experiment is applied by mild heating (47–50 °C), which is not physiological. The implementation of the RBC membrane stabilization model for anti-inflammatory screening is based on the structural similarity between RBC and lysosome membranes. Lysosomes are important organelles that elicit the effectors mechanism in a late inflammatory response through releasing bactericidal enzymes and various proteinases, thereby maintaining their structural integrity is important for suppressing the inflammatory process [43].

Ten out of 18 studies on the anti-inflammatory activity of *Sargassum*’s CSP utilized tropical species [97,99,101,105,107,108,109,111,112,113]. Preetha and Devaraj [99] reported that subcutaneous administration of *S*. *wightii* CSP improved high cholesterol diet-induced hypercholesterolemia in rats, manifested by the improvement of the plasma lipid profile, decreased serum TNF-α, CRP, and plasma fibrinogen, the decreased cardiac mRNA expression of iNOS and COX-2, and the suppression of lysosomal enzymes and cardiac NO production. Fernando et al. [108] studied the effects of CSP from *S*. *polycystum* (SPF) in LPS-induced RAW 264.7 cells. They found that SPF treatment (25–100 µg/mL) reduced NO production, suppressed the expression of iNOS and COX-2 proteins, and increased cell viability. Furthermore, SPF reduced the production of PGE_2_ and some pro-inflammatory cytokines, including TNF-α, IL-1β, and IL-6. Lavanya et al. [105] observed the effect of CSP from *S*. *ilicifolium* (SIF) in TPA-induced polymorphonuclear leukocytes (PMNL). They reported that the SIF treatment increased the viability of PMNL cells and reduced the production of cathepsin D (lysosomal enzyme), nitrite, and TNF-α. Information on the effects of SP from tropical *Sargassum* on inflammatory conditions is more profound than that in the previous section (the crude extract). This is because some researchers have tried to uncover the anti-inflammatory effects on cell units, and have used diverse inflammatory parameters to explain the effects between parameters in improving inflammatory conditions.

Alongside the trends occurring in crude extract screening studies, studies on the anti-inflammatory activity of *Sargassum* purified compounds are still dominated by subtropical samples [19,27,65,108,116,117,118,119,120,121,122,123,124,125,126,127,128,129,130,131,132,133,134,137,140,141,142,143,145,147]. Studies on the anti-inflammatory activity of tropical *Sargassum* purified compounds were carried out by Fernando et al. [26], Maneesh and Chakraborty [136,144], Sarithakumari et al. [138], and Sarithakumari and Kurup [139]. The tested pure compounds of tropical *Sargassum* were fucosterol, pure fucoidan/sulfated polygalactopyranosyl-fucopyranan, aryl polyketide lactones, and alginic acid. Maneesh and Chakraborty [136,144] used an inhibition model of 5-LOX, COX-2, and COX-1 enzymes. Sarithakumari et al. [138] and Sarithakumari and Kurup [139] utilized the rat arthritis model to study the anti-inflammatory activity of alginic acid. Fernando et al. [26] reported that fucosterol isolated from *S*. *binderi* showed satisfactory anti-inflammatory activity in China fine dust particular matter (CPM)-induced A459 human lung epithelial cells, indicated by the suppression of COX-2 activity, decreased production of TNF-α, IL-6, and PGE_2_, inhibition of NF-kB nuclear translocation, and inhibition of MAPK protein phosphorylation. The opportunity to discover the pure bioactive compounds from tropical *Sargassum* is still open, both for the type of unprecedented active compounds and the type of model to be selected, especially the model that can comprehensively explain inflammatory modulation.

### 3.2. Potential Anti-Inflammatory Compounds of Tropical *Sargassum*

Based on the literature summaries in Table 1, Table 2, Table 3 and Table 4, polar and non-polar compounds of both tropical and subtropical *Sargassum* showed promising anti-inflammatory activity in various screening models. Various non-polar compounds from brown seaweed, which are thought to be responsible anti-inflammatory agents include sterol/terpenoids, omega-3 PUFAs, fucoxanthin and its derivatives, grasshopper ketone, and the polyketide group, while the water-soluble anti-inflammatory agents from *Sargassum* are sulfated polysaccharides, alginic acid, β-glucan, and phenolic components.

#### 3.2.1. Lipid-Soluble Bioactive Compounds

The total lipid content in brown seaweed (the Sargassaceae family) is commonly higher in subtropical species (−5% db) than in tropical species (0.8%–1.9% db) [15]. However, high lipid levels (−8% db) were found in tropical *S*. *wightii* harvested from Tamil Nadu, Gulf of Mannar, India [179]. The PUFA content and n3:n6 ratio were also found to be higher in subtropical *Sargassum* [14,150,180]. This could be related to environmental temperature differences. Low-temperature conditions in the subtropical region, especially in winter and fall, can cause an increase of PUFA concentrations in cell membranes, thereby implying the continuity of metabolic processes [181]. Although the PUFA content and n3:n6 ratio in tropical species are lower than those in subtropical ones, the n6:n3 ratio of tropical *Sargassum* still meets World Health Organization (WHO) recommendations (i.e., below 10). Maintaining its ratio in the recommended range can lower the human body’s risk of inflammation-associated diseases, such as cardiovascular and neurological diseases.

Susanto et al. [14] showed that the fucoxanthin level of tropical *Sargassum* (1.64 ± 0.49 mg/g db) was not significantly different from that of subtropical *Sargassum* (1.99–2.12 mg/g db), although there was a higher amount in subtropical *Sargassum*. The fucoxanthin content of seaweed is strongly influenced by species, season, and maturity level [32,182,183]. Lann et al. [184] hypothesized that fucoxanthin in tropical Sargassaceae not only acts as an accessory light-harvesting pigment but also as a photoprotectant. High UV intensity in the tropical area will trigger the formation of free radicals in plant cells, so a certain defense mechanism is required to maintain cell structure integrity.

There is a lack of information on the effects of different latitudes (for tropical and subtropical areas) on the sterol content of *Sargassum*. Terasaki et al. [181] reported that *S*. *horneri* fucosterol content was higher in winter and in deep water seaweed. This phenomenon is related to the seaweed’s response in preserving cell membrane integrity at low temperatures and in suboptimal photosynthetic conditions. Fucosterol plays an important role in supporting liquid-ordered membrane phases and determining membrane integrity by balancing the liquid and non-liquid phases in plant cell membranes [185]. The content of fucosterol in the *S. horneri* harvested from some Japanese waters was in the range of 1.21%–3.21% db [181], while Fleury et al. [186] showed that the *S*. *furcatum* fucosterol content harvested from the water area of Rio De Janeiro, Brazil only reached 0.08% wt.

#### 3.2.2. Water-Soluble Bioactive Compounds

Polyphenol content is generally higher in subtropical seaweeds than in tropical ones [184]. However, some studies have proven the opposite [187,188]. Bolser and Hay [189] revealed that tropical seaweed is likely to form stronger chemical defenses to withstand the intense grazing activity of tropical herbivores. Tropical herbivores are known to be more active and diverse. Bolser and Hay [189] hypothesized that the strong chemical defense of seaweed would be manifested by a high content of secondary metabolites, including those of the polyphenol group. Based on these facts, latitude difference is not the only predictor that determines seaweed’s phenolic content. Various factors affecting the phenolic content in seaweed are age and maturity, the surrounding ecosystem conditions, latitude, geomorphology, depth, irradiation, hydrodynamic conditions, seasons, and water nutrient levels [184]. Tropical *Sargassum* species, especially those growing on the water surface, are rich in low Mw phenolic content (<2 kDa), which is attributed to their role as photoprotectants in plant cells [184]. Audibert et al. [190] showed that the phenolic fraction of brown seaweed *Ascophyllum nodosum* with Mw < 2 kDa had the most potent radical scavenging (ABTS assay) activity compared to the crude phenolic extract and other fractions with higher Mw, although its TPC (total phenolic content) was the lowest among the other fractions. This result indicates that the qualitative composition of the fraction determines its antioxidant activity. Audibert et al. [190] added that the Trolox equivalent antioxidant capacity/TEAC (mmol/l) of the fraction <2 kDa is 3.5 times more active than phloroglucinol and twice more active than quercetin. The antioxidant activity of an active component is often reported to be positively correlated with its anti-inflammatory activity [191,192,193,194].

The biological activity of sulfated polysaccharides (SP) from tropical *Sargassum* has been discussed more frequently than the activity of other compounds. This is due to the high yield potency of SP, which is essential for its commercial development. The FCSP yield of tropical *Sargassum* ranges from 2.16% to 7.15% db [195,196,197]. This reported yield is higher than that of other anti-inflammatory compounds, such as sterols, fucoxanthin, omega-3 PUFAs, and phenolic components. Sulfated polysaccharides are also recognized as potential antioxidants [136,195,198]. Several factors affecting the FCSP content in seaweed include mechanical stress (the level of ocean current and waves), the level of maturity, season, salinity, and part of the thallus [199]. In addition to FCSPs, alginic acid has been reported to show promising anti-inflammatory activity [138,139]. Kokilam et al. [200] reported that the yield of alginic acid in *S. wightii* harvested from Gulf of Mannar, India (tropical) reached 21.71%. Moreover, Aponte de Otaola et al. [201] found that the alginic acid content of several tropical *Sargassum* species (whole plant) originating from Puerto Rico area were in the range of 17.9%–20.3%.

FCSPs and alginates are important components of the wall structures of algae cells. Both of these components play an important role in attaining the flexibility requirements for intertidal seaweed [201,202]. So et al. [203] compared the protective activity of fucoidan and alginic acid against oxidative stress, both in vitro and cellular systems. The results revealed that fucoidan showed stronger scavenging activity for NO and superoxide anions than that of alginic acid. Fucoidan was also reported to be more effective in inhibiting lipid peroxidation in APPH (2,2′-azobis(2-amidinopropane) dihydrochloride)-induced LLK-PK1 cells.

## 4. Anti-Inflammatory Mechanisms of Bioactive Compounds of *Sargassum*

Inflammation is characterized by some general parameters, such as the overproduction of pro-inflammatory cytokines (generally TNF-α, IL-1β, and IL6), NO, PGE_2_, and ROS. Increased NO production occurs along with the increased activity of the iNOS enzyme. This enzyme converts L-arginine to NO [11], while PGE_2_ is produced through cyclooxygenase-2 (COX-2) enzyme activity, using arachidonic acid as a substrate. Arachidonic acid (AA) is a dominant constituent of the phospholipid bilayer in cell membranes. AA is produced from phospholipase A2′s activity against phospholipids. In addition to prostaglandin, the AA substrate can also be converted into pro-inflammatory leukotrienes by the lipoxygenase enzyme (LOX) [31]. *Sargassum* bioactive compounds have been reported to modulate inflammatory responses via inhibition of NF-kB transactivation, inhibition of the MAPK group’s phosphorylation (which impacts the blocking of AP-1 protein translocation), direct NO scavenging, and direct inhibition of important pro-inflammatory enzymes, such as iNOS, COX-2, 5-LOX, and PLA2. 

As discussed in the previous section, NF-kB and AP-1 are important proteins that regulate the transcription of various inflammatory-related genes. The inhibition of their activation suppresses inflammatory response. The downstream responses due to NF-kB and AP-1 transactivation are depicted in Figure 7. Adhesion molecules, such as ICAM-1 and VCAM-1, are typical markers of the vascular system’s inflammation. These compounds help monocytes attach to vascular endothelial cells and promote the initial phase of atherogenesis. Chemoattractant compounds, such as MCP-1, are produced by inflamed cells to recruit various immune cells such as monocytes, macrophages, neutrophils, and eosinophils to the inflammation site [123]. The matrix metalloproteinase (MMP) plays a role in the regulation of transmigration, extravasation, and monocyte infiltration into inflammatory sites through extracellular matrix degradation [204]. The risk of cardiovascular disease could also be characterized by the increased production of fibrinogen. Fibrinogen is known as a coagulation factor protein and is involved in atherosclerosis processes [99]. Sarithakumari et al. [138] used various parameters to describe the inflammatory response in arthritic rats, and one of these parameters was the xanthine oxidase (XO) enzyme. XO activity is associated with increased oxidative stress in cells. This enzyme is responsible for producing superoxide anion radicals and increasing lipid peroxidation [205].

The inhibition of NF-kB activation by *Sargassum* bioactive compounds (not only purified compounds, but also crude extracts and CSP) is proven by decreased phosphorylated NF-kB protein expression (p65 and/or p50), both in the nucleus and in cytoplasmic cell extracts [26,37,75,123,132,135], as well as inhibition of phosphorylation and degradation of the IkB protein in the cytoplasmic extracts [96,100,116,123,125,126,129,147]. The Western blotting and ELISA methods are commonly used to see the expression of several of the aforementioned proteins. Confocal microscopy can also be used to visualize the expression of p-NF-kB through the immunofluorescence technique [129]. An NF-kB-dependent luciferase activity assay is able to monitor NF-kB binding into the DNA binding site [83,132]. Activation of the MAPK protein is usually indicated by levels of the phosphorylated MAPK protein (p38, JNK, and/or ERK1/2) in the cytoplasmic cell extract [26,69,73,75,100,103,116,118,127,128,129,132,135,147] or by an AP-1-dependent luciferase activity assay [132]. Besides inhibiting NF-kB and MAPK group activation, the *Sargassum* bioactive compound (ethanolic extract of *S*. *serratifolium*) is reported to suppress PI3K (phosphatidylinositol 3 kinase) and Akt (protein kinase B) phosphorylation in IL-1β-treated SW1353 chondrocytes. This phenomenon also relates to the improved inflammatory condition [68].

The modulation mechanism of the NF-kB signaling pathway used by *Sargassum* bioactive compounds is not yet clear. The presence of α-β-unsaturated carbonyl groups in terpenoids is thought to be responsible, in part, for NF-kB inhibition because the methylene group can react with the sulfhydryl cysteine groups (Cys^38^ and Cys^120^) located in the DNA binding domain of the p65 monomer (subunit NF-kB) [206]. Heras and Hortelano [153] summarized some specific mechanisms used in modulating the NF-kB signaling pathway by various terpenoid compounds: (1) Inhibition of IkB kinase (IKK) complex activation, (2) inhibition of proteasomal degradation on the IkB protein, and (3) blocking NF-kB nuclear translocation and/or NF-kB binding to the kB site. The most effective approach in NF-kB modulation is inhibition of the IKK complex’s activation.

Some studies have revealed that *Sargassum* bioactive compounds can directly inhibit some pro-inflammatory enzymes, such as COX-2, COX-1, 5-LOX, PLA2, and hyaluronidase, via in vitro techniques [39,136,144]. The existence of electronegative functional groups, such as hydroxyl and methoxy, contained in *Sargassum* bioactive compounds is thought to be responsible for preventing the hydrogen abstraction from arachidonic acid in COX enzymes. Electronegative functionality also has the potential to coordinate the active site of COX/LOX enzymes through ion pairing, thereby preventing the biosynthesis of eicosanoid derivatives [39,136]. Electron-rich structures found in the macrolactone group, such as cyclic esters, the methyl acetate side chain, and vinyl or aryl substituents in the lactone ring, are reported to be positively correlated with the inhibition capacity on the COX-1, COX-2, and 5-LOX enzymes [144]. The reduction of NO concentration in inflamed cells or organisms could be caused by modulation of the upstream NO signaling pathway or direct NO scavenging by antioxidant compounds. Lopes et al. [143] reported that phlorotannin isolated from various species of brown seaweed, including *S*. *vulgare*, showed strong direct scavenging activity of Sodium nitroprusside (SNP)-generated NO in a cell-free system. This fact might contribute to the overall anti-inflammatory effect of phlorotannin in LPS-induced RAW 264.7 cells. Besides several aforementioned modulation mechanisms for *Sargassum* bioactive compounds, Chen et al. [207] proposed that maintaining the equilibrium state of the histone acetylation process as one type of post-translational modification can improve inflammation conditions. They reported that the *Sargassum* sp. polysaccharide extract treatment significantly inhibited the production of inflammatory cytokines and HAT (histone acetyltransferase) activity, but increased HDAC (histone deacetylase) activity and the mRNA expression of HDAC1 in porcine circovirus type 2 (PCV2)-infected RAW 264.7 cells.

Several other manifestations of the *Sargassum*’s bioactive compounds against inflammatory conditions are include elevated intrinsic antioxidant activity (catalase, superoxide dismutase, glutathione peroxidase, and glutathione reductase) [24,123,138], decreased production of ROS [24,33,142], decreased pro-inflammatory cell population (T-helper, T-cytotoxic, granulocytes, eosinophils, and monocytes) [76], stabilization of the lysosomal membrane (which inhibits lysosomal enzymes, such as cathepsin D, glucosaminidase, glucuronidase, etc.) [99,105,139], decreased edema volume and neutrophil infiltration in rat edema model [104], decreased production of osteoclastogenic factors (such as RANKL and osteoprotegerin in the IL-1β-induced MG-63 osteoblast cells) [147], improvement of dyslipidemia conditions [50,99], and amelioration of insulin resistance [33,44]. Increased intrinsic antioxidant enzyme activity is associated with the increased expression of nuclear factor erythroid 2-related factor 2 (Nrf2) transcription factors, the responsible protein in cell defense mechanisms [19,20,77,208]. Lee et al. [209] found that the oral administration of fucosterol (30 mg/kg/ day for seven days) in CCl_4_-induced rat hepatotoxicity was able to improve the nuclear translocation of Nrf2. Related to the improvement of dyslipidemia, Preetha and Devaraj [99] hypothesized that CSP treatment in hypercholesterol diet-induced rats could delay cholesterol absorption in the intestine and might accelerate cholesterol excretion by modulating cholesterol-7-alpha-hydroxylase, a key enzyme for the conversion of cholesterol into bile acids.

Another mechanism that may be ascribed to the anti-inflammatory activity of *Sargassum* bioactive compounds is the interaction between active compounds and inflammatory-associated receptors, such as TLR4, CD14, CR-3, and SR-A. This interaction can interfere with the recognition of strong inflammatory stimuli by cells, e.g., LPS. The presence of sulfate groups in FCSPs is thought to affect its binding to inflammatory receptors [169,210]. Omega-3 fatty acids (especially SDA and EPA) in *Sargassum* may undergo incorporation into cell membranes and affect eicosanoid metabolism in cells. EPA and DHA can interfere with the activity of the COX-2 and 5-LOX enzymes against AA, so the production of strong pro-inflammatory PGE_2_ and LT_4_ can be inhibited [211]. Tian et al. [212] found that RAW 264.7 cell incubation with EPA, DHA, or DPA (docosapentaenoic acid) for 72 h suppressed the inflammatory response after LPS stimulation.

## 5. Conclusions

*Sargassum* is well-known to have beneficial effects on health and is traditionally utilized by the East Asian community to treat various inflammatory-related diseases. Various bioactive compounds of *Sargassum* show potential anti-inflammatory activity, both in acute and chronic conditions. The potency of tropical *Sargassum* as a source of the anti-inflammatory agent is not fully explored yet. Sulfated polysaccharides from tropical *Sargassum* are the most studied compounds for their anti-inflammatory activity. However, there are various other potentially bioactive compounds, such as non-polar compounds (fatty acids and derivatives, carotenoids and derivatives, steroids, and polar lipids), grasshopper ketone, polyketide macrolactone, and phenolic components. Bioactive compounds of *Sargassum* are reported to modulate the inflammatory response by inhibiting NF-kB and MAPK activation, direct inhibition of inflammation-associated enzymes, and direct scavenging of radical species. The information presented in this paper is expected to provide input for *Sargassum* development, especially its tropical species, as a source of anti-inflammatory agents.

## Figures and Tables

**Figure 1 marinedrugs-17-00590-f001:**
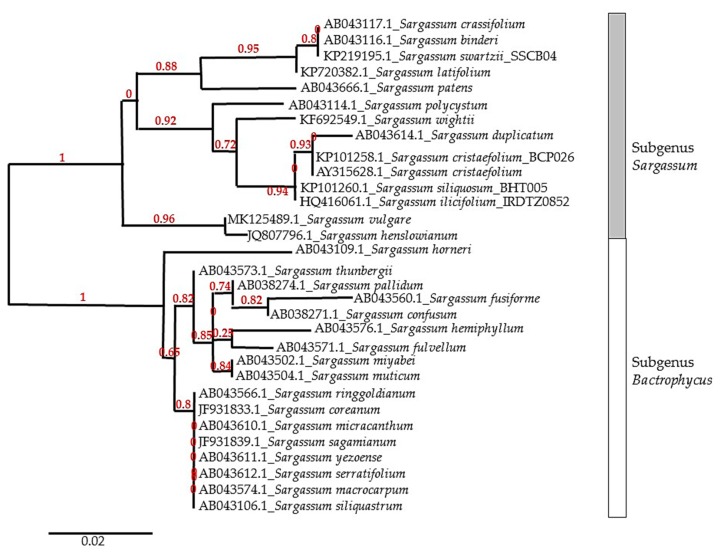
Phylogenetic relationship of genus *Sargassum*, implemented by the maximum likelihood method based on internal transcribed spacer (ITS-2) gene alignment. The numbers at each node represent the bootstrap value.

**Figure 2 marinedrugs-17-00590-f002:**
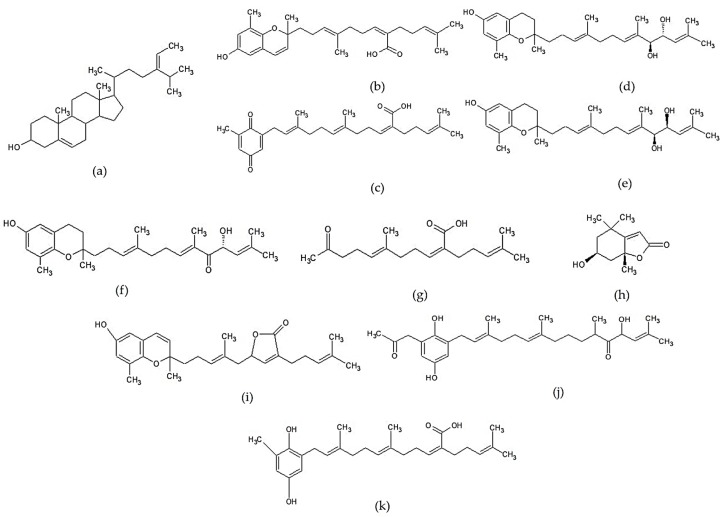
Chemical structures of several terpenoid compounds of *Sargassum* sp.: (**a**) fucosterol [156], (**b**) sargachromenol [126], (**c**) sargaquinoic acid [126], (**d**) sargachromanol D [157], (**e**) sargachromanol E [158], (**f**) sargachromanol G [159], (**g**) isoketochabrolic acid [125], (**h**) loliolide [160], (**i**) tuberatolide B [161], (**j**) isonahocol E3 [128] “Reprinted from European Journal of Pharmachology, 720/1-3, Sah et al., Novel isonahocol E3 exhibits anti-inflammatory and anti-angiogenic effects in endothelin-1-stimulated human keratinocytes, 205-211, Copyright (2013), with permission from Elsevier”, and (**k**) sargahydroquinoic acid [162].

**Figure 3 marinedrugs-17-00590-f003:**
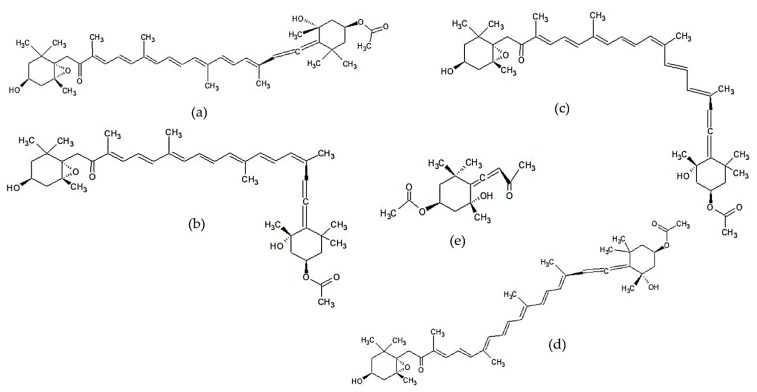
The chemical structures of fucoxanthin and its derivatives of *Sargassum* sp.: (**a**) all-trans-(6′R)-fucoxanthin [165], (**b**) 9-cis-(6′R)-fucoxanthin [117], (**c**) 13′-cis-(6′R)-fucoxanthin [117], (**d**) 13-cis-(6′R)-fucoxanthin [117] “Reprinted from Food and Chemical Toxicology, 50/9, Heo et al., Anti-inflammatory effect of fucoxanthin derivatives isolated from *Sargassum siliquastrum* in lipopolysaccharide-stimulated RAW 264.7 macrophage, 3336-3342, Copyright (2012), with permission from Elsevier”, and (**e**) apo-9′-fucoxanthinone [166].

**Figure 4 marinedrugs-17-00590-f004:**
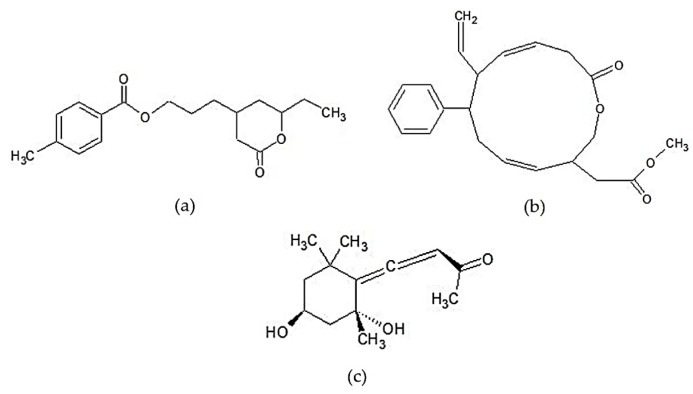
Chemical structures of other lipid-soluble compounds of *Sargassum* sp.: (**a**) 4-(8-ethyl-tetrahydro-7-oxo-2H-pyran-5-yl)-propyl-4’-methyl benzoate, (**b**) methyl-2-(12-oxo-7-phenyl-8-vinyl-1-oxa-4,9- cyclododecadien-3-yl)-acetate [144] “Reprinted from Food Research International, 100, Maneesh A. and Chakraborty K., Unprecedented antioxidative and anti-inflammatory aryl polyketides from the brown seaweed *Sargassum wightii*, 640-649, Copyright (2017), with permission from Elsevier”, and (**c**) grasshopper ketone [168].

**Figure 5 marinedrugs-17-00590-f005:**
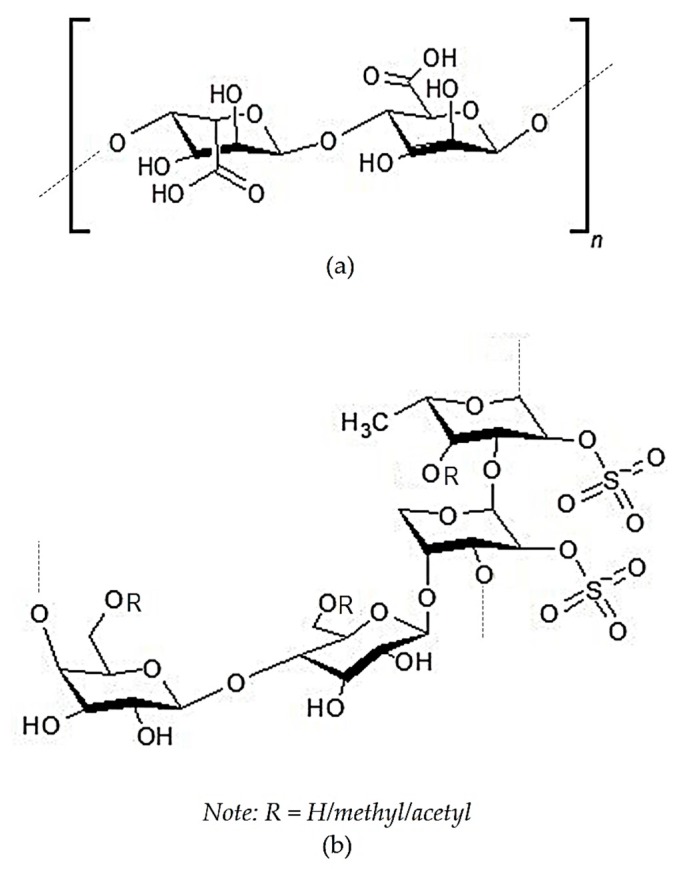
Chemical structures of the polysaccharide compounds of *Sargassum* sp.: (**a**) Alginic acid [172] and (**b**) sulfated polygalactopyanosil fucopyranan (an example of purified fucose-containing sulfated polysaccharides (FCSPs)) [136] “Reprinted by permission from [Springer Nature Customer Service Centre GmbH]: [Springer Nature] [Journal of Applied Phycology] [Pharmacological potential of sulfated polygalactopyranosyl-fucopyranan from the brown seaweed *Sargassum wightii*, Maneesh A. And Chakraborty K), [COPYRIGHT] (2018)”.

**Figure 6 marinedrugs-17-00590-f006:**
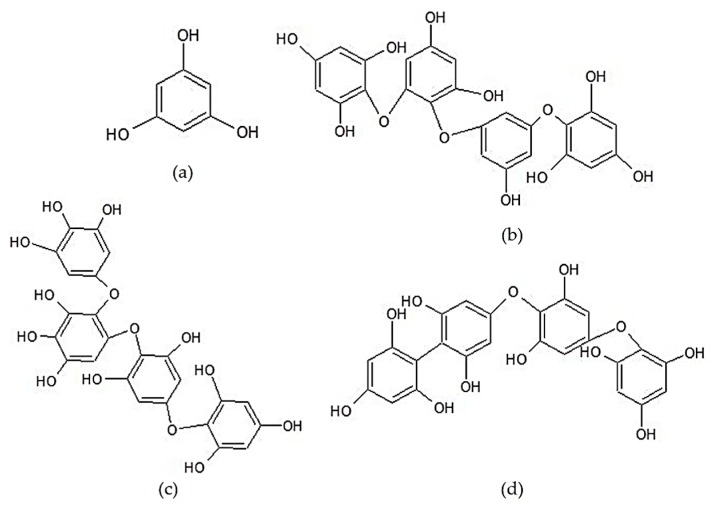
Chemical structures of the phenolic compounds of *Sargassum* sp.: (**a**) Phloroglucinol (a monomer of phlorotannin), (**b**) tetraphloretol B (an example of the phlorethol group), (**c**) tetrafuhalol A (an example of the fuhalol group), and (**d**) fucodiphlorethol A (an example of the fucophlorethol group) [143].

**Figure 7 marinedrugs-17-00590-f007:**
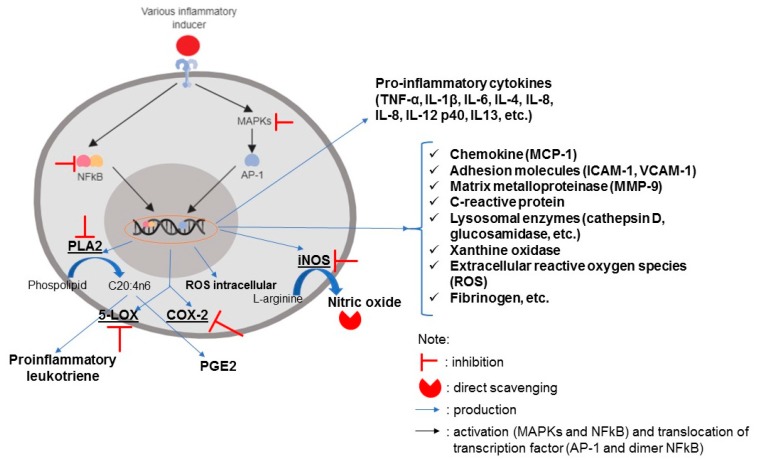
The inflammatory signaling pathway and the resulting downstream responses. Signs of and show the potential modulation of *Sargassum* bioactive compounds on the inflammatory response, including (1) the inhibition of NF-kB and MAPK activation, (2) the inhibition of pro-inflammatory enzymes PLA2, 5-LOX, COX-2, iNOS, and (3) the direct scavenging of radical species. This illustration was created by the author using Biorender.com.

**Table 1 marinedrugs-17-00590-t001:** Studies on the anti-inflammatory activity of *Sargassum* crude extracts and their partitions.

Sample	Observed Response	Tested Compound	Model	Ref.
**Subgenus *Sargassum***				
*S*. *binderi*	(1) Decreased serum interleukin (IL)-2, tumor necrosis factor (TNF)-α, and IL-1β production, improve wound healing rate, and elevated serum IL-4 ^1^ (2) Decreased nitric oxide (NO) production ^2,3^	Phyto-oleic acid nanovesicles (PONVs) made by petroleum ether extract ^1^; water extract ^2^; and ethanolic precipitate of water extract ^3^	Wounded streptozotocin-induced diabetic rats ^1^; and lipopolysaccharide (LPS)-induced RAW 264.7 ^2,3^	^1^ [34] * ^2^ [35] * ^3^ [36] *
*S*. *patens*	(3) Decreased NO, prostaglandin E_2_ (PGE_2_), IL-6, TNF-α, inducible nitric oxide synthase (iNOS) and cyclooxygenase (COX)-2 production ^1,2^ (4) Inhibition of p65 nuclear factor kappa B (NF-kB) translocation ^1^ (5) Decreased ear edema volume and mastocyte infiltration ^1^	Ethanol extract ^1,2^	LPS-induced RAW 264.7 ^1,2^; and croton oil-induced rat ear edema ^1^	^1^ [37] ^2^ [38]
*S*. *wightii*	(1) Significant inhibition of 5-lipoxygenase (LOX), COX-1, and COX-2 in vitro ^1^ (2) Inhibition of in vitro albumin denaturation ^2,6^ (3) Decreased paw edema volume ^3,4,5^ (4) Inhibition of red blood cells (RBC) hemolysis and proteinase activity ^6^	Methanol-ethyl acetate extract ^1^; chloroform extract ^1,3^; ethanol extract ^2,3^; hexane extract ^3, 6^; methanol extract ^4,5,6^; butanol extract ^4^; and ethyl-acetate extract ^6^	In vitro inhibition of 5-LOX, COX-1, and COX-2 ^1^; albumin denaturation inhibition ^2,6^; carrageenan-induced rat paw edema ^3,4,5^; and RBC membrane stabilization and proteinase inhibition ^6^	^1^ [39] * ^2^ [40] * ^3^ [41] * ^4^ [23] ^5^ [42] * ^6^ [43] *
*S*. *swartzii*	(1) Decreased serum TNF-α, IL-6, and C-reactive protein (CRP) production in diabetic rats ^1^ (2) Decreased paw edema volume and inflammatory exudate ^2^ (3) Reduction of inflammatory cells accumulation, cell swelling, and dilated sinusoids in hepatocytes through histopathological observation ^3^ (4) Decreased NO production ^4^	Gold nanoparticles of water extract ^1^; methanol extract ^2^; water extract ^3^; and diethyl ether fraction of methanol extract ^4^	Alloxan-induced diabetic Wistar rats ^1^; carrageenan-induced rat paw edema and peritonitis ^2^; acetaminophen-induced hepatotoxicity in mice ^3^; and LPS-induced RAW 264.7 ^4^	^1^ [44] * ^2^ [45] * ^3^ [46] ^4^ [47] *
*S*. *crassifolium*	Decreased rat paw edema volume	β-glucan extract yielded from acid and ultrasound methods	Carrageenan-induced rat (*Rattus novergicus*) paw edema	[48] *
*S. ilicifolium*	Decreased paw edema volume	Methanol extract;	Carrageenan-induced rat paw edema	[49] *
*S. duplicatum*	Decreased NO production ^1,2^	Water extract ^1^; and ethanolic precipitate of water extract ^2^	LPS-induced RAW 264.7 ^1,2^	^1^ [35] * ^2^ [36] *
*S. subrepandum*	Improved plasma lipid profile (plasma cholesterol, triglycerides, low density lipoprotein/LDL, and high density lipoprotein/HDL), decreased serum malondialdehyde (MDA), NO, TNF-α, and leptin production, and increased serum adiponectin level	Methanol extract (100 mg/kg)	Atherogenic diet-induced female Sprague Dawley rats	[50]
*S. polycystum*	(1) Decreased rat paw edema volume ^1^ (2) Decreased serum TNF-α and intact condition of hepatocytes ^2^ (3) Improvement of inflammatory condition in histopathological evaluation of liver and kidney tissue ^3^ (4) Decreased production of NO and proinflammatory cytokines (TNF-α, IL-1β, and IL-6) ^4^	Hexane fraction from methanol extract ^1^; ethanol extract ^2,3^; water extract ^3^; hexane, dichloromethane, and methanol extract ^4^	Carrageenan-induced rat paw edema ^1^; acetaminophen-induced hepatoxicity in rat ^2^; high calorie diet and low dose streptozotocin-induced type II diabetes ^3^; and LPS-induced C8B4 microglia cells ^4^	^1^ [51] * ^2^ [52] * ^3^ [53] * ^4^ [54] *
*S. vulgare*	(1) Decreased rat paw edema volume ^1^ (2) Inhibition of COX-1 and COX-2 activity ^2^	Methanol extract ^1^; ethyl acetate extract ^2^	Carrageenan-induced rat paw edema ^1^; and in vitro inhibition to COX-1 and COX-2 enzymes ^2^	^1^ [55] ^2^ [56]
**Subgenus *Bactrophycus***				
*S. thunbergii*	(1) Decreased production of NO, TNF-α ^1,2,4^, IL-1β, and IL-6 ^1,2^ (2) Decreased ear edema volume ^3^ (3) Suppression of iNOS expression ^4^ (4) Decreased NO production and increased reactive oxygen species (ROS) scavenging activity ^5^ (5) Suppression of matrix metalloproteinase (MMP)-2 and MMP-9 expression ^6^ (6) Suppression of ear edema and erythema ^7^ (7) Decreased production of myeloperoxidase (MPO) and enhanced production of antioxidant enzymes (superoxide dismutase (SOD)-1 and glutathion reductase) ^8^	Ethanolic extract of fermented samples and its fractions ^1^; ethyl acetate extract ^2^; extract of dichloromethane, ethanol, and water ^3^; ethanol 70% extract ^4^; ethanol extract ^5^; and methanol extract ^6,7,8^	LPS-induced RAW 264.7 ^1,2,5^; rat ear edema ^3^; LPS-induced BV-2 microglial cells ^4^; H_2_O_2_-induced RAW 264.7 ^5^; phorbol 12-myristate 13 acetate (PMA)-induced HT1080 ^6^; PMA-induced mouse ear edema and erythema ^7^; and TNF-α stimulated human monocytic leukemia ^8^	^1^ [18] ^2^ [57] ^3^ [22] ^4^ [58] ^5^ [59] ^6^ [60] ^7^ [61] ^8^ [62]
*S. fulvellum*	(1) Decreased production of COX-2, iNOS, pro-inflammatory cytokines, NO, and PGE_2_ ^1,2,3^ (2) Increased expression of CU/Zn- superoxide dismutase (SOD) and reduced ROS production ^1^ (3) Decreased ear edema volume ^4^ (4) Suppressed NO production ^5,6^ and reduced cytoplasmic activity ^5,^ (5) Decreased serum IL-4, immunoglobulin E (IgE), and TNF-*α* production, and reduction of IL-4, IL-5, and IL-13 in mice splenocytes ^7^ (6) Suppression of ear edema and erythema formation ^8^	Ethyl acetate fraction from ethanol extract ^1^, hexane fraction ^2^; ethanol extract ^3,4,7^; dichloromethane extract ^4^; and water extract ^4,5^; ethanol precipitate of water extract ^6^; and methanol extract ^8^	Ultraviolet B (UVB)-induced HaCaT keratinocytes and BALB/c mice ^1^; LPS-induced RAW 264.7 ^2,3,5, 6^; rat ear edema ^3,4^; dinitrochlorobenzene (DNCB)-induced atopic dermatitis (AD)-like skin lesions in BALB/c mice ^7^; and PMA-induced mouse ear edema and erythema ^8^	^1^ [24] ^2^ [63] ^3^ [64] ^4^ [22] ^5^ [35] * ^6^ [36] * ^7^ [65] ^8^ [61]
*S. serratifolium*	(1) Decreased NO, PGE_2_, iNOS, and COX-2 production ^1,2,4^ (2) Inhibition of inhibitor kappa B (IkB) degradation ^1,3^ (3) Decreased expression of intercellular adhesion molecule (ICAM)-1, vascular cell adhesion molecule (VCAM)-1, monocyte chemoattractant protein (MCP)-1, keratinocyte chemoattractant (KC) ^3,6^, and MMP-9 ^6^, and elevated heme oxygenase 1 expression ^3^ (4) Inhibition of NF-kB transactivation ^3,4,6^, MAPKs phosphorylation, and phosphoinositide 3 kinase/protein kinase B (PI3K/Akt) activation ^4^ (5) Decreased expression of MMP-1, MMP-3, and MMP-13 ^4^ (6) Decreased macrophage infiltration and MCP-1 expression in epididymal tissue ^5^	Ethanol extract ^1,2,4^; hexane fraction from ethanolic extract ^3^; meroterpenoid-rich extract from hexane fraction ^5,6^	LPS-induced mouse peritoneal macrophage ^1^; LPS-induced BV-2 microglial cells and LPS-induced rat hippocampus cells ^2^; TNF-α-induced human umbilical vein endothelial cells (HUVECs) ^3,6^; IL-1β-treated SW1353 human chondrocytes ^4^; high fat (HF)-fed C57BL/6J mice ^5^; and high cholesterol diet (HCD)-fed C57BL/6J mice ^6^	^1^ [66] ^2^ [67] ^3^ [20] ^4^ [68] ^5^ [28] ^6^ [29]
*S. miyabei*	(1) Suppressed NO, IL-6, TNF-α, IL-1β, iNOS, and COX-2 production (2) Decreased mitogen-activated protein kinases (MAPKs) and NF-kB activation (3) Decreased ear edema volume	Ethanol extract	LPS-induced RAW 264.7 and rat ear edema	[69]
*S. fusiforme*	(1) Suppressed ear edema formation and RBL (rat basophilic leukemia) degranulation ^1^ (2) Decreased activity of phospholipase A2 (PLA2), COX-2, LOX, and hyaluronidase ^1^ (3) Decreased NO ^2,4^ and ROS ^4^ production (4) Decreased TNF-α production, and elevated production of IL-6 and IL-10 production ^3^ (5) Decreased NO, iNOS, COX-2, and pro-inflammatory cytokines production via inhibition of MAPKs activation ^5^ (6) Suppression of ear edema and erythema formation ^6^	Administration of diethyl ether fraction percutaneously and orally ^1^; ethyl acetate fraction from ethanol 30% extract ^2^; water extract ^3^; methanol extract and its fractions (dichloromethane, ethyl acetate, n-butanol, water) ^4^; ethanol extract of fermented and non-fermented sample ^5^; and methanol extract ^6^	Rat ear edema and RBL cells ^1^, LPS-induced RAW 264.7 ^2,4,5^; tert-butyl hydroperoxide (t-BHP)-induced RAW 264.7 RAW 264.7 ^4^; LPS-induced C2C12 myotube cells ^3^; and PMA-induced mouse ear edema and erythema ^6^	^1^ [31] ^2^ [70] ^3^ [71] ^4^ [33] ^5^ [72] ^6^ [61]
*S. horneri*	(1) Decreased production of NO ^1,2,4,6,7,8,9^, PGE_2_ ^2,4,6,7,9^, and pro-inflammatory enzymes (iNOS and COX-2) ^1,4,6,7,9^ (2) Inhibition of extracellular signal-regulated kinase (ERK) ^1,4,6,9^, c-Jun-N-terminal kinase (JNK) ^4,6,9^, p38 ^1,6,9^, and NF-κB ^1,4,6,9^ activation (3) Hindered cell migration and decreased MMP-2 and MMP-9 activity ^3^ (4) Decreased population of T helper, T cytotoxic, granulocytes, eosinophil, and monocytes ^5^ (5) Decreased production of TNF-α ^5,6,7^ IL-1β ^5,6,7^, IL-6 ^5,6,7^, IL-4 ^5^, IL-5 ^5^, and IL-13 ^5^ (6) Elevated expression of hemeoxygenase 1 ^8^ (7) Decreased NO production and increased ROS scavenging activity ^10^ (8) Suppression of ear edema and erythema formation ^11^	Ethanol 70% extract ^1,4,5^; ethanol 85% extract ^2^; ethanol extract ^3,6,7,10^ and its fraction ^3,6^; methanol 80% extract and its fraction ^8^; combination of *Ecklonia cava* and S. *horneri* ethanol 70% extracts ^9^; and methanol extract ^11^	LPS-induced RAW 264.7 ^1,2,4,6,8,9,10^; PMA-induced HT1080 fibrosarcoma ^3^; concanavalin A-induced rat splenocytes ^5^; fine dust (FD)-induced RAW 264.7 ^7^; and PMA-induced mouse ear edema and erythema ^11^	^1^ [73] ^2^ [74] ^3^ [30] ^4^ [75] ^5^ [76] ^6^ [19] ^7^ [77] ^8^ [78] ^9^ [79] ^10^ [59] ^11^ [61]
*S. coreanum*	(1) Decreased production of NO, IL-6, TNF-α, IL-1β, iNOS, and COX-2 via inhibition of NF-kB activation (2) Decreased ear edema volume	Ethanol extract	LPS-induced RAW 264.7 and rat ear edema	[80]
*S. ringgoldianum*	Suppression of mouse ear edema and erythema formation	Methanol extract	PMA-induced mouse ear edema and erythema	[61]
*S. micracanthum*	(1) Decreased production of NO, IL-6, TNF-α, IL-1β, iNOS, and COX-2 ^1^ (2) Inhibition of NF-kB activation ^1^ (3) Decreased ear edema volume ^2^	Water extract ^1^; and ethanol extract ^2^	LPS-induced RAW 264.7 ^1^; and croton oil-induced rat ear edema ^2^	^1^ [81] ^2^ [82]
*S. macrocarpum*	(1) Decreased production of IL-12 p40, IL-6, and TNF-α (2) Inhibition of NF-kB nuclear translocation and IkB degradation ^1,2^ (3) Decreased production of NO, PGE2, iNOS, COX-2, TNF-α, and IL-1β ^2^ (4) Suppression of PI3K/Akt and ERK phosphorylation and increased nuclear factor erythroid-2-related factor 2 (Nrf2) and hemeoxygenase-1 expression ^2^	Ethanol 70% extract ^1,2^	CpG-DNA-induced bone marrow-derived macrophages (BMDMs) and bone marrow-derived dendritic cells (BMDCs) from C57BL/6 mice ^1^; and LPS-induced RAW 264.7 ^2^	^1^ [83] ^2^ [84]
*S. sagamianum*	(1) Decreased production of NO, IL-6, IL-1β, TNF-α, iNOS, and COX -2 via inhibition of NF-kB p65 activation ^1^ (2) Decreased ear edema volume ^1^ (3) Suppression of mouse ear edema and erythema formation ^2^	Ethanol extract ^1^; and methanol extract ^2^	LPS-induced RAW 264.7 and rat ear edema ^1^; and PMA-induced mouse ear edema and erythema ^2^	^1^ [85] ^2^ [61]
*S. muticum*	(1) Decreased NO and PGE_2_ production, iNOS and COX-2 expression (protein and mRNA), IL-1β and IL-6 expression (mRNA) ^1^ (2) Decreased production of IL-6, TNF-α, and interferon (IFN)-γ in serum and lymphocyte ^2^ (3) Improved arthritis score, edema condition, and histology of the knee joint ^2^ (4) Suppressed MMP-1 expression via inhibition of activator protein (AP)-1 activation and its binding to MMP-1 promoter ^3^ (5) Decreased production of serum IL-2, TNF-α, and IL-1β ^4^ (6) Suppression of p38 and JNK phosphorylation ^5^	Ethanol 80% extract and its fraction (hexane, CH_2_Cl_2_, EtOAc, BuOH, and water) ^1^; ethanol 70% extract ^2^; ethyl acetate fraction of ethanol extract ^3,5^; and methanol extract ^4^	LPS-induced RAW 264.7 ^1^; Collagen-induced arthritis DBA/1J mice ^2^; UVB-induced HaCaT keratinocytes ^3,5^; and STZ-induced hepatic injury in Wistar rats ^4^	^1^ [21] ^2^ [27] ^3^ [86] ^4^ [87] ^5^ [88]
*S. hemiphyllum*	(1) Decreased production of histamine and β-hexosaminidase in PMA-induced rat peritoneal mastocyte ^1^ (2) Decreased production of IL-8 and TNF-α in A23187-induced HMC-1 ^1^ (3) Inhibition of NF-kB activation in TNF-α-induced 293T cells ^1^ (4) Decreased iNOS expression via inhibition of MAPKs activation ^2^ (5) Decreased NO production and increased ROS scavenging activity ^3^	Methanol extract ^1,2^; and ethanol extract ^3^	PMA-induced rat peritoneal mastocyte, A23187-induced HMC-1/human mast cell, and TNF-α-induced 293T cells ^1^; β-amyloid protein (Aβ)-induced HT-22 mouse neuronal cells ^2^; and LPS and H_2_O_2_-induced RAW 264.7 ^3^	^1^ [89] ^2^ [90] ^3^ [59]
*S. confusum*	(1) Decreased iNOS expression via inhibition of MAPKs activation ^1^ (2) Suppression of mouse ear edema and erythema formation ^2^	Methanol extract ^1,2^	β-amyloid protein (Aβ)-induced HT-22 mouse neuronal cells ^1^; and PMA-induced mouse ear edema and erythema ^2^	^1^ [90] ^2^ [61]
*S. siliquastrum*	Suppression of NO and iNOS production	Aqueous extract of sample fermented by *Lactobacillus* sp. SH-1	LPS-induced RAW 264.7	[91]
*S. pallidum*	(1) Enhanced production of serum IL-2, IL-4, and IL-10 (2) Decreased production of serum IL-6, IL-1β, and TNF-α	Water extract	N-methyl-N′-nitro-nitrosoguanidine (MNNG)-induced gastric cancer rats	[92]

Note: References followed by an asterisk (*) used samples from a tropical area. The superscripted numbering of reference, model, tested compund, and observed response in the same row are related to one another. This superscripted numbering is discontinous between rows and only applied to the same row.

**Table 2 marinedrugs-17-00590-t002:** Studies on the anti-inflammatory activity of *Sargassum* crude sulfated polysaccharides (CSP) and their fractions.

Sample	Observed Response	Tested Compound	Model	Ref
**Subgenus *Sargassum***				
*S. wightii*	(1) Suppressed edema formation, neutrophil migration, and peritoneal exudate production ^1,2^ (2) Improved serum lipid profile, and decreased TNF-α, CRP, fibrinogen, iNOS, NO, COX-2, and lysosomal enzymes production ^3^	CSP ^2,3^; and fraction of CSP (Fr IV) ^1^	Carrageenan-induced rat paw edema, carrageenan-induced peritonitis, and Freund’s adjuvant-induced arthritis ^1,2^; and hypercholesterol diet-induced rat dyslipidemia ^3^	^1^ [101] * ^2^ [107] * ^3^ [99] *
*S. cristaefolium*	(1) Decreased production of NO and iNOS (2) Inhibition of p38, JNK, ERK, and NF-kB activation	CSP and its fraction (1193.2, 864.4, 386.1 kDa, 55.9, 15.4, and 1.9 kDa)	LPS-induced RAW 264.7	[103]
*S. ilicifolium*	(1) Increased PMNL viability (2) Decreased production of cathepsin D, nitrate, and TNF-α.	CSP	TPA (12- O- Techanoyl 13—Myristate)-induced polymorphonuclear leukocytes (PMNL)	[105] *
*S. asperifolium*	Decreased production of NO and TNF-α	CSP	LPS-induced lymph macrophage	[106]
*S. vulgare*	Suppressed paw edema formation	CSP	Carrageenan-induced rat paw edema	[97] *
*S. polycystum*	Decreased production of NO ^1,2^, PGE_2_, TNF-α, IL-1β, and IL-6 ^1^	CSP ^1,2^	LPS-induced RAW 264.7	^1^ [108] * ^2^ [109] *
*S. latifolium*	Decreased production of NO, TNF-α, and COX-2	Different fraction of water-soluble polysaccharide extracts (not only sulfated form)	LPS-induced RAW 264.7	[110]
*S. siliquosum*	(1) Decreased production of TNF-α, IL-1β, IL-6 ^1,2^, and MCP-1 ^2^ (2) Decreased NO production	Different fraction of water-soluble polysaccharide extracts (not only sulfated form) ^1,2,3^	LPS-induced peripheral blood mononuclear cells ^1,2^; and LPS-stimulated promyelocyticleukemic cells ^3^	^1^ [111] * ^2^ [112] * ^3^ [113] *
**Subgenus *Bactrophycus***				
*S. horneri*	(1) Decreased production of NO, PGE_2_, and pro-inflammatory cytokines ^1,2,3^ (2) Inhibition of NF-kB activation ^1,2^ and phosphorylation of p-38 and ERK1/2 ^2^ (3) Decreased NO production in LPS-induced zebra fish ^1^	Fraction of CSP yielded from membrane filtration (<5 kDa (f1), 5–10 kDa (f2), 10–30 kDa (f3), and >30 kDa (f4)) ^1^; and CSP resulted from Celluclast enzyme digestion ^2^, CSP and its fractions (Q Sepharose Fast Flow column) ^3^	LPS-induced RAW 264.7 ^1,2,3^; and LPS-induced zebra fish ^1^	^1^ [96] ^2^ [100] ^3^ [102]
*S. hemiphyllum*	(1) Hindered formation of erythema and ear edema, and decreased neutrophil infiltration ^1^ (2) Decreased production of MPO, NO, IL-1β, IL-6, and TNF-α in rat ear^1^ and RAW 264.7 ^2^	CSP	Arachidonic acid-induced rat ear edema ^1^; and LPS-induced RAW 264.7 ^2^	^1^ [104] ^2^ [98]

Note: References followed by an asterisk (*) used samples from tropical area. The superscripted numbering of reference, model, tested compund, and observed response in the same row are related to one another. This superscripted numbering is discontinous between rows and only applied to the same row.

**Table 3 marinedrugs-17-00590-t003:** The anti-inflammatory activity of *Sargassum* purified compounds and their mechanisms in modulating inflammation (lipid-soluble compounds).

Compound	Source	Modulation of Inflammation	Ref
**Terpenoid group**			
Fucosterol	*S. binderi*	Suppression of COX-2, PGE2, TNF-α, and IL-6 production via the inhibition of NF-kB activation and MAPK group phosphorylation	[26]*
Sargachromenol	*S. serratifolium*	Suppression of adhesion molecules (VCAM-1, and ICAM-1) and chemotactic cytokine (MCP-1) production via inhibition of IKK-β - Ikβ phosphorylation, and NF-kB nuclear translocation in TNF-α-induced HUVECs	[123]
	*S. micracanthum*	Suppression of pro-inflammatory cytokines (TNF-α, IL-1β, and IL-6), PGE2, NO, COX-2, and iNOS production via inhibition of Ikβ degradation in LPS-induced RAW 264.7	[126]
	*S. horneri*	Suppression of MMP-1, -2, and -9 via inhibition of AP-1 activation (c-Jun and c-Fos) in UVA-induced human derman fibroblast	[120]
	*S. macrocarpum*	Inhibition of JNK and ERK phosphorylation and increased ROS scavenging activity in UVB-induced HaCaT keratinocytes	[121]
Sargaquinoic acid	*S. serratifolium*	Suppression of adhesion molecules (VCAM-1, and ICAM-1) and chemotactic cytokine (MCP-1, and IL-8) production via inhibition of Ikβ degradation in TNF-α-induced HUVECs	[124]
	*S. siliquastrum*	Suppression of iNOS and NO production via inhibition of Ikβ degradation, NF-kB nuclear translocation, and JNK1/2 phosphorylation in LPS-induced RAW 264.7	[129]
Sargahydroquinoic acid	*S. yezoense*	Suppression of MMP-2/-9 expression via inhibition of NF-kB nuclear translocation, Ikβ degradation, and AP-1 activation in TNF-α stimulated HaCaT cells	[119]
Sargachromanol D	*S. siliquastrum*	Suppression of pro-inflammatory cytokines (TNF-α, IL-1β, and IL-6), PGE2, NO, COX-2, and iNOS production via inhibition of p65 and Ikβ-α phosphorylation in LPS-induced RAW 264.7	[127]
Sargachromanol E	*S. siliquastrum*	Suppression of pro-inflammatory cytokines (TNF-α, IL-1β, and IL-6), PGE2, NO, COX-2, and iNOS production via inhibition of MAPKs group phosphorylation (JNK, ERK, and p38) LPS-induced RAW 264.7	[118]
Sargachromanol G	*S. siliquastrum*	Suppression of pro-inflammatory cytokines (TNF-α, IL-1β, and IL-6), PGE2, NO, COX-2, and iNOS production via inhibition of IkB-α, NF-κB (p65 and p50), and MAPK (ERK1/2, JNK, and p38) phosphorylation in LPS-induced RAW 264.7	[116]
		Suppression of osteoclastogenic factor (PGE2, COX-2, IL-6, OPG, and RANKL) via inhibition of IkB-α, NF-κB (p65 and p50), and MAPKs (ERK1/2, JNK, and p38) phosphorylation in IL-1β-induced MG-63 osteoblast cells	[147]
Isoketochabrolic acid (IKCA)	*S. micracanthum*	Suppression of pro-inflammatory cytokines (TNF-α, IL-6, and IL-1β), PGE2, NO, COX-2, and iNOS production in LPS-induced RAW 264.7	[125]
Tuberatolide B	*S. macrocarpum*	Suppression of NO, PGE2, IL-6, IL-1β, iNOS, and COX-2 production via inhibition of NF-κB (p65) and MAPK (ERK1/2, JNK, and p38) phosphorylation, and IkB degradation LPS-induced RAW 264.7	[122]
Isonahocol E3	*S. siliquastrum*	Suppression of IL-6, IL-8, and TNF-α production, and MMP gene expression via inhibition of ERK phosphorylation in ET-1-induced human keratinocytes	[128]
Loliolide	*S. horneri*	Suppression of NO production in LPS-induced RAW 264.7	[19]
**Carotenoid group**			
Fucoxanthin	*S. siliquastrum*	Suppression of pro-inflammatory cytokines (TNF-α, IL-1β, and IL-6), PGE2, NO, COX-2, and iNOS production in LPS-induced RAW 264.7	[117]
Apo-9′-fucoxanthinone	*S. muticum*	Suppression of NO ^1,2^, PGE2, proinflammatory cytokines (TNF-α, IL-6, and IL-1β), iNOS, and COX-2 production via inhibition of NF-κB (p65) and MAPK (ERK1/2, JNK, and p38) phosphorylation, and IkB degradation in LPS-induced RAW 264.7 ^1^	[130]^1^; [27]^2^
		Suppression of NO and PGE2 production via inhibition of Ikβ degradation in LPS-induced RAW 264.7	[133]
		Suppression of pro-inflammatory cytokines (IL-12 p40, TNF-α, and IL-6) and iNOS production via inhibition of ERK phosphorylation and AP-1 translocation in CpG DNA-induced BMDMs (bone marrow-derived macrophages) and BMDC (bone marrow-derived dendritic cells)	[132]
		Suppression of IgE, IL-4, interferon- gamma, and TNF-α production, and lymph node size in atopic dermatitis rats	[131]
**Other group**			
Aryl polyketide lactone	*S. wightii*	Direct inhibition of 5-LOX, COX-2, and COX-1 enzymes (in vitro)	[144] *
Grasshopper ketone	*S. fulvellum*	Suppression of pro-inflammatory cytokines (TNF-α, IL-1β, and IL-6), NO, COX-2, and iNOS production via inhibition of p65 NF-κB nuclear translocation and MAPK (ERK1/2, JNK, and p38) phosphorylation in LPS-induced RAW 264.7 Suppression of IFN-γ and IL-4 production in concanavalin-A-induced BALB/c mice splenocytes	[145] [65]

Note: References followed by an asterisk (*) used samples from tropical area. The superscripted numbering as listed in reference and modulation of inflammation in the same row are related to one another. This superscripted numbering is discontinous between rows and only applied to the same row.

**Table 4 marinedrugs-17-00590-t004:** Anti-inflammatory activity of *Sargassum* purified compounds and their mechanism in modulating inflammation (water-soluble compounds).

Compound	Source	Modulation of Inflammation	Ref
**Polysaccharides**			
Purified FCSPs (fucoidan)	*S. wightii*	Direct inhibition of 5-LOX, COX-2, and COX-1 enzymes (in vitro).	[136] *
	*S. henslowianum*	Increased secretion of anti-inflammatory cytokines (IL-2, IL-4, and IL-10) and suppression of pro-inflammatory cytokines (IL-6 and TNF-α) production in MNNG-induced gastric cancer rats.	[134]
	*S. hemiphyllum*	A combination of oligofucoidan (LMF) and fucoxanthin resulted in enhancement of the intestinal epithelial barrier and immune function against LPS stimulation through suppression of IL-1β and TNF-α production and increased secretion of IL-10 and IFN-γ in CaCo2 cells co-cultured with *B*. *lactis* ^1^ A combination of oligofucoidan (LMF) and fucoxanthin resulted in enhancement of adiponectin production, and decreased production of TNF-α and IL-6 in type II diabetes mouse model ^2^	[137]^1^ [141]^2^
	*S. horneri*	Decreased NO production (IC50 = 40 μg/mL) via inhibition of the NF-κB and MAPK (ERK and p38) signaling pathways in LPS-stimulated RAW 264.7 cells. Decreased heart-beating rate, cell death, ROS, and NO levels in LPS-exposed zebrafish embryos.	[140]
Alginic acid	*S. horneri*	Suppression of PGE2, proinflammatory cytokines (TNF-α, IL-6, and IL-1β), and COX-2 production via inhibition of NF-κB (p65) nuclear translocation and MAPK (ERK1/2, JNK, and p38) phosphorylation in CFD (Chinese fine dust)-induced HaCaT keratinocytes.	[135]
	*S. wightii*	Suppression of COX-2, 5-LOX, MPO, xanthine oxidase (XO), ceruloplasmin, rheumatoid factor, CRP, pro-inflammatory cytokines, and lysosomal enzymes in type-2 collagen-induced rat arthritis.	[139] *
		Suppression of COX-2, 5-LOX, MPO, XO, ceruloplasmin, rheumatoid factor, and CRP production, and enhancement of antioxidant enzymes activity in Freund’s complete adjuvant-induced rat arthritis.	[138] *
**Phenolic compounds**			
Phlorotannin	*S. muticum*	Suppression of ROS production in PMA-induced neutrophil and suppression of PGE2, COX-1, and COX-2 expression in A23187-induced erythrocytes.	[142]
	*S. vulgare*	Suppression of NO production in LPS-induced RAW 264.7 and direct scavenging of NO in a cell-free system.	[143]

Note: References followed by an asterisk (*) used samples from tropical area. The superscripted numbering as listed in reference and modulation of inflammation in the same row are related to one another. This superscripted numbering is discontinous between rows and only applied to the same row.

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
