# Peer review of "Sargassum Seaweed as a Source of Anti-Inflammatory Substances and the Potential Insight of the Tropical Species: A Review"

_marinedrugs, 2019, doi:10.3390/md17100590_

Round 1

Reviewer 1 Report

Sargassum is generally well-written and will be of interest to the readers of Marine Drugs. However, it has a major flaw in its construction, which must be addressed. The article differentiates between tropical and subtropical species of Sargassum. Although there is some explanation of which species of Sargassum are considered tropical or subtropical is given at the beginning of section 3.1, this is insufficient. It will aid the reader if an earlier and more comprehensive explanation of which species are considered tropical and subtropical is given. Mark in each table, clearly what are tropical and subtropical species. A phylogenetic tree, again with tropical and subtropical species, could further assist the reader.
Lines 96-97 and 334-340 are unclear.

Author Response

Hope you can find this response well. Firstly, I would like to say thanks for your comments and suggestions. We really appreciate to your input in order to improve the manuscript writing quality.

Here are some responses to your comments:

Issue #1: Tropical and subtropical species differentiation, and phylogenetic tree issue

In this revised manuscript, we provide a brief introduction about Sargassum subgenus differentiation in line of 97-113. Further explanation about tropical and subtropical differentiation can be found in line of 386-399. Actually, most of tropical Sargassum used in anti-inflammatory studies originate from Subgenus Sargassum. But we found that some species of Sargassum subgenus were also harvested in subtropical area as explained in line of 393-395. On the contrary, although almost all of subtropical species used in anti-inflammatory studies originate from Subgenus Bactrophycus, we also found that one species (S. fulvellum) of Bactrophycus subgenus  was harvested from Malaysian waters (citation No. 40 and 45). According to those facts, in order to aid the readers to differentiate the samples, we made a differentiation between Subgenus Sargassum and Bactrophycus in Table 1 and 2, and we added an  asterisk to references which use tropical Sargassum as samples in anti-inflammatory study (we put this note after Table “References followed by an asterisk (*) used samples from tropical area”).

Line 97-113

“Various Sargassum species used in several studies of anti-inflammatory activity originated from two subgenera of Sargassum, namely Bactrophycus and Sargassum. The Sargassum genus is generally divided into four subgenera: Phyllotrichia, Bactrophycus, Arthrophycus and Sargassum. This subgenus classification is based on the morphological characteristics of the seaweed thallus. In addition, the distribution pattern can also be used to distinguish between Sargassum subgenera. The most common subgenera found in subtropical/temperate regions are Bactrophycus and Arthrophycus. Subgenus Phyllotrichia is only found in Australia and adjacent areas, while the Sargassum subgenus is widely distributed in  tropical areas [16]. Due to the high level of morphological plasticity caused by differences in environmental condition, molecular marker techniques combined with morphological observation are implemented to resolve taxonomic issues [17]. A sequence of the internal transcribed spacer of nuclear ribosomal DNA (ITS) is commonly used to analyze the phylogenetic relationship among Sargassum species. Figure 1 shows the phylogenetic tree of the Sargassum genus based on internal transcribed spacer (ITS)-2 gene sequences. The phylogenetic tree was constructed based on several Sargassum species used in various anti-inflammatory activity studies. The accession number of each gene sequences is obtained from the database of National Center for Biotechnology Information.”

Line 386-399

“The majority of Sargassum crude extracts tested on anti-inflammatory screening are derived from subtropical samples. Seventeen out of 73 studies used samples from tropical regions [35,38,53,55–59,177,40,44,45,48–52]. Most of the observed tropical species came from subgenus Sargassum, including S. polycyctum, S. wightii, S. swartzii, S. crassifolium, S. binderi, and S. ilicifolium. Some subtropical species tested in the crude extract studies were dominated by the subgenus Bactrophycus, including S. hemyphyllum, S. muticum, S. sagamianum, S. macrocarpum, S. micracanthum, S. coreanum, S. horneri, S. fusiforme, S. miyabei, S. serratifolium, S. fulvellum, S. confusum, S. siliquastrum, S. pallidum, S. ringgoldianum, and S. thunbergii. However, some species belonging to the subgenus Sargassum can also be found in subtropical areas, such as S. patens, S. wightii, S. vulgare, S. subrepandum, and S. swartzii [31,43,46,47,54,60,61]. The differentiation between tropical and subtropical samples is based on the thorough evaluation of the sampling location or coordinates information provided in each study. Tropical samples originated from area near the equator (from 23.5° further north to 23.5° southern latitude). While subtropical samples originated from area between 23.5° and 66.5° north and south.”

Issue #2: An unclear sentence in the line of 96-97

This sentence of “Mun et al. [15] compared the ethanolic extract’s anti-inflammatory activity of fermented brown seaweed S. thunbergii and unfermented one in lipopolysaccharide (LPS)-stimulated RAW 264.7 macrophage cells...” has been changed to be “Mun et al. [18] compared the anti-inflammatory activity of the ethanolic extracts from fermented and non-fermented (fresh) samples of the brown seaweed S. thunbergii in lipopolysaccharide (LPS)-stimulated RAW 264.7 macrophage cells. Fermentation was initiated in fresh S. thunbergii by inoculation of kimchii-isolated Lactobacillus sp. SH-1.....”

Issue #3: An unclear explanation in the line of 334-340

This issue is related to the issue of sample differentiation (issue #1)

Issue #4: Addition of new references

According to the recommendation of one reviewer, we tried to re-evaluate of the literature using several literature-search engines to ensure that all primary literatures related to the anti-inflammatory study of Sargassum species are included to this manuscript (the literature searching was conducted until August, 8th 2019, so the recent paper published after that date may be not included). After the re-evaluation process, we found 36 new papers (primary literatures) and citations of new papers in the manuscript are yellow-highlighted. Details of new citations are as follows: 1 new citation in the introduction section (Husni et al. [10]), 34 new citations in the section 2, and 1 new citation in the section 4 (Chen et al. [209]).

From this re-evaluation procedures, we found that we have evidently missed two anti-inflammatory compounds isolated from Sargassum, namely sargahydroquinoic acid (citation No. 128) and grasshopper ketone (citation No. 70 and 146). We have included those two compounds to this revised manuscript and provided their chemical structures. Besides this issue, we added a short explanation about another possible mechanism of Sargassum bioactive compound in modulating inflammatory response as proven by Chen et al. [209].

Hopefully the addition of some new papers can improve the value of this manuscript.

Issue #5: Written english quality

We have conducted some changes related to the english writing techiques. Our manuscript also has undergone english languange editing by MDPI english editing service. All changes regarding this issue can be seen through “track changes” feature in Ms. Word.

We do apologize for all of our limitation. We are looking forward to see your comment and suggestion about this manuscript. Please see the revised manuscript in this attachment. 

Reviewer 2 Report

Sargassum seaweed as a source of anti-inflammatory substances and the potential insight of the tropical species: a review

Saraswati, Puspo Edi Giriwono, Diah Iskandriati, Chin Ping Tan, Nuri Andarwulan

The layout of this review is logical. The information contained in its tables and figures represents a useful compilation of the known literature on tropical Sargassum seaweeds and their anti-inflammatory properties, which is likely to serve as a valuable resource to the Marine Drugs readership.  The written English could be improved with respective to grammar through thorough editing by a native English speaker, mainly focusing on subject / object of sentences and use of prepositions, as well as context-appropriate synonyms.

It is recommended that the manuscript be accepted for publication after this recommended editing.

Examples of the editing required are given for the first three pages in the accompanying document.

Author Response

Hope you can find this response well. Firstly, I would like to say thanks for your comments and suggestions. We really appreciate to your input in order to improve the manuscript writing quality. We have followed your instruction as given in the previous accompanying document. All changes can be seen through “track changes” feature in Ms. Word. We would like to clarify some issues related to the revision procees of this manuscript.

Issue #1: Written english quality

We have conducted some changes related to the english writing techiques. Our manuscript also has undergone english languange editing by MDPI english editing service.

Issue #2: Addition of new references

According to the recommendation of one reviewer, we tried to re-evaluate of the literature using several literature-search engines to ensure that all primary literatures related to the anti-inflammatory study of Sargassum species are included to this manuscript (the literature searching was conducted until August, 8th 2019, so the recent paper published after that date may be not included). After the re-evaluation process, we found 36 new papers (primary literatures) and citations of new papers in this manuscript are yellow-highlighted. Details of new citations are as follows: 1 new citation in the introduction section (Husni et al. [10]), 34 new citations in the section 2, and 1 new citation in the section 4 (Chen et al. [209]).

From this re-evaluation procedures, we found that we have evidently missed two anti-inflammatory compounds isolated from Sargassum, namely sargahydroquinoic acid (citation No. 128) and grasshopper ketone (citation No. 70 and 146). We have included those two compounds to this revised manuscript and provided their chemical structures. Besides this issue, we added a short explanation about another possible mechanism of Sargassum bioactive compound in modulating inflammatory response as proven by Chen et al. [209].

Hopefully the addition of some new papers can improve the value of this manuscript.

Issue #3: Manuscript construction regarding tropical-subtropical species differentiation and phylogenetic tree issue

One reviewer told us to address the issue related to the manuscript construction. This manuscript actually tries to give special insight about tropical Sargassum potency as an anti-inflammatory agent,  although the potential of subtropical species are also not ignored. But one of reviewer highlighted that the issue of tropical-subtropical species differentiation become fundamental problem that will affect the quality of manuscript construction. We were recommended to give an earlier and more comprehensive explanation about tropical-subtropical species differentiation and to add phylogenetic tree which may further assist the reader.

In this revised manuscript, we provide a brief introduction about Sargassum subgenus differentiation in line of 97-113. Further explanation about tropical and subtropical differentiation can be found in line of 386-399. Actually, most of tropical Sargassum used in anti-inflammatory studies originate from Subgenus Sargassum. But we found that some species of Sargassum subgenus were also harvested in subtropical area as explained in line of 393-395. On the contrary, although almost all of subtropical species used in anti-inflammatory studies originate from Subgenus Bactrophycus, we also found that one species (S. fulvellum) of Bactrophycus subgenus  was harvested from Malaysian waters (citation No. 40 and 45). According to those facts, in order to aid the readers to differentiate the samples, we made a differentiation between Subgenus Sargassum and Bactrophycus in Table 1 and 2, and we added an  asterisk to references which use tropical Sargassum as samples in anti-inflammatory study (we put this note after Table “References followed by an asterisk (*) used samples from tropical area”).

Line 97-113

“Various Sargassum species used in several studies of anti-inflammatory activity originated from two subgenera of Sargassum, namely Bactrophycus and Sargassum.

The Sargassum genus is generally divided into four subgenera: Phyllotrichia, Bactrophycus, Arthrophycus and Sargassum. This subgenus classification is based on the morphological characteristics of the seaweed thallus. In addition, the distribution pattern can also be used to distinguish between Sargassum subgenera. The most common subgenera found in subtropical/temperate regions are Bactrophycus and Arthrophycus. Subgenus Phyllotrichia is only found in Australia and adjacent areas, while the Sargassum subgenus is widely distributed in  tropical areas [16]. Due to the high level of morphological plasticity caused by differences in environmental condition, molecular marker techniques combined with morphological observation are implemented to resolve taxonomic issues [17]. A sequence of the internal transcribed spacer of nuclear ribosomal DNA (ITS) is commonly used to analyze the phylogenetic relationship among Sargassum species. Figure 1 shows the phylogenetic tree of the Sargassum genus based on internal transcribed spacer (ITS)-2 gene sequences. The phylogenetic tree was constructed based on several Sargassum species used in various anti-inflammatory activity studies. The accession number of each gene sequences is obtained from the database of National Center for Biotechnology Information.”

Line 386-399

“The majority of Sargassum crude extracts tested on anti-inflammatory screening are derived from subtropical samples. Seventeen out of 73 studies used samples from tropical regions [35,38,53,55–59,177,40,44,45,48–52]. Most of the observed tropical species came from subgenus Sargassum, including S. polycyctum, S. wightii, S. swartzii, S. crassifolium, S. binderi, and S. ilicifolium. Some subtropical species tested in the crude extract studies were dominated by the subgenus Bactrophycus, including S. hemyphyllum, S. muticum, S. sagamianum, S. macrocarpum, S. micracanthum, S. coreanum, S. horneri, S. fusiforme, S. miyabei, S. serratifolium, S. fulvellum, S. confusum, S. siliquastrum, S. pallidum, S. ringgoldianum, and S. thunbergii. However, some species belonging to the subgenus Sargassum can also be found in subtropical areas, such as S. patens, S. wightii, S. vulgare, S. subrepandum, and S. swartzii [31,43,46,47,54,60,61]. The differentiation between tropical and subtropical samples is based on the thorough evaluation of the sampling location or coordinates information provided in each study. Tropical samples originated from area near the equator (from 23.5° further north to 23.5° southern latitude). While subtropical samples originated from area between 23.5° and 66.5° north and south.”

Please see the revised manuscript in the attachment file. We do apologize for all of our limitation. We are looking forward to see your comment and suggestion regarding this manuscript. 

Reviewer 3 Report

The review evaluates recent studies the potential of sargassum seaweed as anti-inflammatory source of natural products. Unfortunately, a few issues render the manuscript difficult to follow. 

References are not up to date:

At least 5 of the 6 references reviewing the topic are not included. I suggest a thorough evaluation of the literature and inclusion of these very thorough references (please use PUBMED, SCIFINDER and google scholar). In fact, I found those paper quite informative, and some areas redundant with your efforts. 

A very good recent paper was not included.                       

Prev Nutr Food Sci. 2019 Jun;24(2):150-158. doi: 10.3746/pnf.2019.24.2.150. Epub  2019 Jun 30.

Some of the use of language is colloquial and flow of the article is poorly designed.

example: "The most popular active polysaccharides derived from brown seaweed are FCSPs/fucoidan and alginic acid." From a scientific perspective the sentence makes no sense, what is meant by the most popular. I do not understand the context, but a more appropriate sentence would be: The most abundant polysaccharide compounds found in brown seaweed are FCSPs (define the word) such as fucoidan and alginic acid. 

Even thought the information might be useful to the community, its writing style and difficult format obscures its readability.

Author Response

Hope you can find this response well. Firstly, I would like to say thanks for your comments and suggestions. We really appreciate to your input in order to improve the manuscript writing quality. We would like to clarify some issues related to the revision procees of this manuscript.

Issue #1: Addition of new references due to unupdated references

In the revision process, we tried to re-evaluate of the literature using several literature-search engines to ensure that all primary literatures related to the anti-inflammatory study of Sargassum species are included to this manuscript (the literature searching was conducted until August, 8th 2019, so the recent paper published after that date may be not included). After the re-evaluation process, we found 36 new papers (primary literatures) and citations of new papers in the manuscript are yellow-highlighted. Details of new citations are as follows: 1 new citation in the introduction section (Husni et al. [10]), 34 new citations in the section 2, and 1 new citation in the section 4 (Chen et al. [209]).

From this re-evaluation procedures, we found that we have evidently missed two anti-inflammatory compounds isolated from Sargassum, namely sargahydroquinoic acid (citation No. 128) and grasshopper ketone (citation No. 70 and 146). We have included those two compounds to this revised manuscript and provided their chemical structures. Besides this issue, we added a short explanation about another possible mechanism of Sargassum bioactive compound in modulating inflammatory response as proven by Chen et al. [209]. We do apologize for this limiation. Your advices is really valuable.

Hopefully the addition of some new papers can improve the value of this manuscript.

Issue #2: Written english quality (bad writing style)

We have conducted some changes related to the writing techiques, by following up some inputs from reviewers. Our manuscript also has undergone english languange editing by MDPI english editing service. All changes regarding this issue can be seen through “track changes” feature in Ms. Word.

Issue #3: Unclear sentence in the line of 281-282

The sentence of “The most popular active polysaccharides derived from brown seaweed are FCSPs/fucoidan and alginic acid...” has been changed to be “The most abundant active polysaccharides derived from brown seaweed are FCSPs and alginic acid...”. Actually FCSPs and alginic acid is two different polysaccharide group. FCSPs stands for fucose-containing sulfated polysaccharides (as defined earlier in the previous session). FCSPs is commonly known as fucoidan. While alginic acid is anionic polysaccharide which does not possess sulfate group in its chemical structure.

Issue #4: Manuscript construction (difficult format)

This manuscript actually tries to give special insight about tropical Sargassum potency as an anti-inflammatory agent sources, although the potential of subtropical species are also not ignored. But one of reviewer highlighted that the issue of tropical-subtropical species differentiation become fundamental problem that will affect the quality of manuscript construction. We were recommended to give an earlier and more comprehensive explanation about tropical-subtropical species differentiation and to add phylogenetic tree which may further assist the reader.

In this revised manuscript, we provide a brief introduction about Sargassum subgenus differentiation in line of 97-113. Further explanation about tropical and subtropical differentiation can be found in line of 386-399. Actually, most of tropical Sargassum used in anti-inflammatory studies originate from Subgenus Sargassum. But we found that some species of Sargassum subgenus were also harvested in subtropical area as explained in line of 393-395. On the contrary, although almost all of subtropical species used in anti-inflammatory studies originate from Subgenus Bactrophycus, we also found that one species (S. fulvellum) of Bactrophycus subgenus  was harvested from Malaysian waters (citation No. 40 and 45). According to those facts, in order to aid the readers to differentiate the samples, we made a differentiation between Subgenus Sargassum and Bactrophycus in Table 1 and 2, and we added an  asterisk to references which use tropical Sargassum as samples in anti-inflammatory study (we put this note after Table “References followed by an asterisk (*) used samples from tropical area”).

Line 97-113

“Various Sargassum species used in several studies of anti-inflammatory activity originated from two subgenera of Sargassum, namely Bactrophycus and Sargassum.

The Sargassum genus is generally divided into four subgenera: Phyllotrichia, Bactrophycus, Arthrophycus and Sargassum. This subgenus classification is based on the morphological characteristics of the seaweed thallus. In addition, the distribution pattern can also be used to distinguish between Sargassum subgenera. The most common subgenera found in subtropical/temperate regions are Bactrophycus and Arthrophycus. Subgenus Phyllotrichia is only found in Australia and adjacent areas, while the Sargassum subgenus is widely distributed in  tropical areas [16]. Due to the high level of morphological plasticity caused by differences in environmental condition, molecular marker techniques combined with morphological observation are implemented to resolve taxonomic issues [17]. A sequence of the internal transcribed spacer of nuclear ribosomal DNA (ITS) is commonly used to analyze the phylogenetic relationship among Sargassum species. Figure 1 shows the phylogenetic tree of the Sargassum genus based on internal transcribed spacer (ITS)-2 gene sequences. The phylogenetic tree was constructed based on several Sargassum species used in various anti-inflammatory activity studies. The accession number of each gene sequences is obtained from the database of National Center for Biotechnology Information.”

Line 386-399

“The majority of Sargassum crude extracts tested on anti-inflammatory screening are derived from subtropical samples. Seventeen out of 73 studies used samples from tropical regions [35,38,53,55–59,177,40,44,45,48–52]. Most of the observed tropical species came from subgenus Sargassum, including S. polycyctum, S. wightii, S. swartzii, S. crassifolium, S. binderi, and S. ilicifolium. Some subtropical species tested in the crude extract studies were dominated by the subgenus Bactrophycus, including S. hemyphyllum, S. muticum, S. sagamianum, S. macrocarpum, S. micracanthum, S. coreanum, S. horneri, S. fusiforme, S. miyabei, S. serratifolium, S. fulvellum, S. confusum, S. siliquastrum, S. pallidum, S. ringgoldianum, and S. thunbergii. However, some species belonging to the subgenus Sargassum can also be found in subtropical areas, such as S. patens, S. wightii, S. vulgare, S. subrepandum, and S. swartzii [31,43,46,47,54,60,61]. The differentiation between tropical and subtropical samples is based on the thorough evaluation of the sampling location or coordinates information provided in each study. Tropical samples originated from area near the equator (from 23.5° further north to 23.5° southern latitude). While subtropical samples originated from area between 23.5° and 66.5° north and south.”

Please see the revised manuscript in the attachment file. We are looking forward to see your comment and suggestion.

Round 2

Reviewer 1 Report

Thank you for response and corrections